# CrossMPT: Cross-Attention Message-Passing Transformer for Error Correcting Codes

**Seong-Joon Park**[1,*]  **Hee-Youl Kwak**[2,†]  **Sang-Hyo Kim**[3]  **Yongjune Kim**[1,†]  **Jong-Seon No**[4]

[1]POSTECH, [2]University of Ulsan, [3]Sungkyunkwan University, [4]Seoul National University
joonpark2247@gmail.com, ghy1228@gmail.com, iamshkim@skku.edu,
yongjune@postech.ac.kr, jsno@snu.ac.kr

## ABSTRACT

Error correcting codes (ECCs) are indispensable for reliable transmission in communication systems. Recent advancements in deep learning have catalyzed the exploration of ECC decoders based on neural networks. Among these, transformer-based neural decoders have achieved state-of-the-art decoding performance. In this paper, we propose a novel Cross-Attention Message-Passing Transformer (CrossMPT), which shares key operational principles with conventional message-passing decoders. While conventional transformer-based decoders employ a self-attention mechanism without distinguishing between magnitude and syndrome embeddings, CrossMPT updates these two types of embeddings separately and iteratively via two masked cross-attention blocks. The mask matrices are determined by the code's parity-check matrix, which explicitly captures and removes irrelevant relationships between the magnitude and syndrome embeddings. Our experimental results show that CrossMPT significantly outperforms existing neural network-based decoders for various code classes. Notably, CrossMPT achieves this decoding performance improvement while significantly reducing memory usage, computational complexity, inference time, and training time.

## 1 INTRODUCTION

The fundamental objective of digital communication systems is to reliably transmit information from source to destination through noisy channels. Error correcting codes (ECCs) are crucial for ensuring the integrity of transmitted data in digital communication systems. The advancements in deep learning across diverse tasks, such as natural language processing (NLP), image classification, and object detection (Devlin et al., 2019; He et al., 2016; Girshick et al., 2014; Carion et al., 2020), have motivated the application of deep learning techniques to ECC decoders. This has led to the development of neural decoders (Kim et al., 2018; 2020; Nachmani et al., 2016; 2018; Dai et al., 2021; Lugosch & Gross, 2017), which aims to improve decoding performance by overcoming limitations of the conventional decoders such as belief propagation (BP) (Richardson & Urbanke, 2001) and min-sum (MS) (Fossorier et al., 1999) decoders.

Among neural decoders, model-free neural decoders employ arbitrary neural network architectures (e.g., deep neural networks (Gruber et al., 2017), recurrent neural networks (Bennatan et al., 2018) and transformers (Choukroun & Wolf, 2022a; 2023; Park et al., 2025; Choukroun & Wolf, 2024a;b)) as the ECC decoder, without relying on prior knowledge of specific decoding algorithms. Since model-free neural decoders are not based on specific decoding algorithms, their training is susceptible to overfitting, largely due to the exponentially large number of possible codewords (Bennatan et al., 2018). To circumvent overfitting, these neural decoders incorporate a preprocessing step where the magnitude and syndrome vectors from the received codewords are concatenated and used as inputs. The preprocessing step is crucial for integrating an effective network architecture for the ECC decoder without an overfitting issue (Bennatan et al., 2018). For instance, transformer-based ECC decoders incorporating this preprocessing (Choukroun & Wolf, 2022a; 2023; Park et al., 2025; Choukroun & Wolf, 2024a;b) achieve state-of-the-art decoding performance for short block codes.

---

*The source code is available at https://github.com/iil-postech/crossmpt.
†Corresponding authors

However, two key questions remain unaddressed: 1) how to effectively process the two distinct input vectors (magnitude and syndrome), and 2) how to design a more efficient transformer-based decoder architecture.

Conventional transformer-based ECC decoders, initially proposed as Error Correction Code Transformer (ECCT) (Choukroun & Wolf, 2022a; 2023; Park et al., 2025; Choukroun & Wolf, 2024a;b), process the concatenated magnitude and syndrome vectors as a single input, applying self-attention blocks without distinguishing between them. In contrast, our approach treats the magnitude and syndrome as *multimodal data*, recognizing their distinct informational characteristics. The *real-valued* magnitude vector represents bit reliabilities, while the *binary* syndrome vector conveys the information of erroneous bit positions. This deliberate separation necessitates the development of a novel architecture, specifically designed to effectively update these separated magnitude and syndrome vectors, thereby significantly improving decoding performance.

In this paper, we introduce a novel Cross-Attention Message-Passing Transformer (CrossMPT) for ECC decoding. CrossMPT processes the magnitude and syndrome embeddings separately to leverage their distinct informational properties. It employs two *cross-attention blocks* to iteratively update the magnitude and syndrome embeddings. Initially, the magnitude embedding is encoded into the *query*, while the syndrome embedding is encoded into *key* and *value*. The first cross-attention block utilizes this configuration in its attention mechanism to update the magnitude embedding. This procedure is reciprocated for the syndrome embedding, which is encoded into the query, while the updated magnitude embedding is encoded into the key and value. This configuration enables the second cross-attention block to update the syndrome embedding. These two masked cross-attention blocks iteratively collaborate to refine the magnitude and syndrome embeddings as in the message-passing algorithm (Richardson & Urbanke, 2001).

To facilitate training, CrossMPT employs a mask matrix for each cross-attention block. The first cross-attention block uses the transpose of the parity check matrix (PCM) $H^T$ as its mask matrix with the magnitude embedding as the query. In the second cross-attention block, the PCM $H$ itself is applied as the mask matrix, with the syndrome embedding as the query. This strategy leverages the PCM's inherent representation of the 'magnitude-syndrome' relationship, aligning with the architecture's objectives. Moreover, the combined size of the two attention maps of CrossMPT is at most half that of conventional transformer-decoders, significantly reducing memory usage. This reduction allows CrossMPT to train and decode longer codes, which previous approaches (concatenating magnitude and syndrome embeddings) are unable to achieve due to high memory usage and computational complexity. To our knowledge, CrossMPT is the first architecture to integrate an iterative message-passing framework with a cross-attention-based transformer decoder.

Experimental results show that CrossMPT consistently outperforms the original ECCT across various code classes. Leveraging its shared operational principles with the message-passing algorithm, CrossMPT demonstrates particularly improved decoding performance, especially for low-density parity-check (LDPC) codes. Notably, we also demonstrate that CrossMPT closely approaches the maximum likelihood decoding performance on short codes. Beyond its enhanced decoding performance, CrossMPT significantly reduces computational costs, including floating point operations (FLOPs), training time, and inference time, compared to the original ECCT. Since the decoder layer constitutes a substantial portion of the total computational cost, this reduction leads to a significant decrease in overall computational complexity.

## 2 RELATED WORKS

In the field of neural network-based ECC decoders, there are two primary categories: the model-based decoder and the model-free decoder. First, model-based decoders are constructed based on the conventional decoding methods (e.g., BP decoder and MS decoder). They map the iterative decoding process of the conventional decoding methods into neural networks and train the network weights accordingly. To improve performance over the standard BP decoder, the recurrent neural network was employed for the decoding of BCH codes (Nachmani et al., 2018). Several recent studies showed that neural network-based BP and MS decoders outperform the conventional decoding algorithms over various code types (Dai et al., 2021; Kwak et al., 2022; 2023; 2025; Lugosch & Gross, 2017; Nachmani & Wolf, 2019; 2021; Buchberger et al., 2021). However, model-based

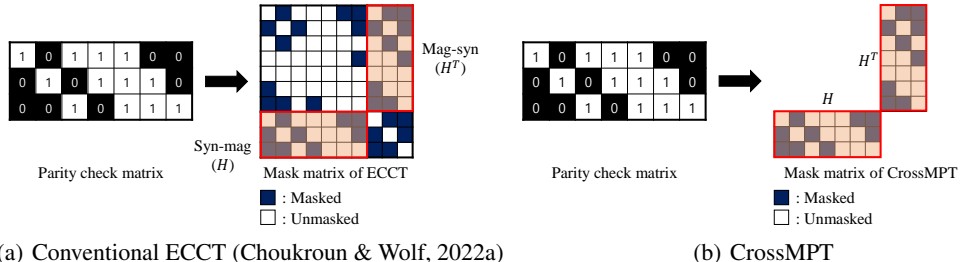

Figure 1: The PCMs and the mask matrices of ECCT and CrossMPT.

neural decoders may encounter performance limitations due to their restrictive model architectures, which are closely tied to underlying decoding methods.

Unlike model-based decoders, which are constrained by the limitations of their underlying algorithms (e.g., BP), model-free neural decoders use arbitrary architectures to learn decoding without such restrictions. While early approaches (Gruber et al., 2017; Cammerer et al., 2017; Kim et al., 2018) employed fully connected networks, they faced overfitting challenges during training. Subsequently, the introduction of a preprocessing step utilizing the magnitude and syndrome vectors of the received codeword to learn multiplicative noise has been pivotal in enabling model-free decoders to address the overfitting issue (Bennatan et al., 2018). Then, ECCT (Choukroun & Wolf, 2022a) first employed the transformer architecture using the same preprocessing step and demonstrated that the transformer-based decoder outperforms existing neural decoders including model-based neural decoders. Building on the ECCT framework, denoising diffusion error correction codes (Choukroun & Wolf, 2023) interpreted the iterative decoding process as a diffusion process and incorporated a diffusion model to train the original ECCT. Recently, multiple-masks ECCT (Park et al., 2025) utilized different PCMs for the same linear code to capture the diverse multilateral relationships of the magnitude and syndrome bits and improve the decoding performance. Notably, transformer-based decoders outperform model-based neural decoders and serve as universal decoders capable of decoding arbitrary code classes with a unified architecture.

Furthermore, employing cross-attention mechanisms in architecture has enhanced performance across various domains. In NLP, the transformer decoder (Vaswani et al., 2017) adopted the cross-attention modules. In vision, CrossViT (Chen et al., 2021) utilized cross-attention for improved image classification. For text-based image generation, latent diffusion models (Rombach et al., 2022) integrated cross-attention layers into the model architecture, enabling diffusion models to become powerful and flexible generators. These works demonstrate the versatility of cross-attention, inspiring its application to ECC decoding.

# 3 BACKGROUND

## 3.1 ERROR CORRECTING CODES

Let $C$ be a linear block code, which is defined by a generator matrix $G$ of size $k \times n$ and a parity check matrix $H$ of size $(n-k) \times n$. They satisfy $GH^\top = 0$ over $\{0,1\}$ with modulo 2 addition. A codeword $x \in C \subset \{0,1\}^n$ is encoded by multiplying message $m$ with the generator matrix $G$ (i.e., $x = mG$). Let $x_s$ be the binary phase shift keying (BPSK) modulated signal of $x$ and let $y$ be the output of a noisy channel for input $x_s$. We assume the additive white Gaussian noise (AWGN) channel and the channel output can be represented by $y = x_s + z$, where $z \sim \mathcal{N}(0, \sigma^2)$. The objective of the decoder is to recover the transmitted codeword $x$ by correcting errors. When $y$ is received, the decoder first determines whether the received signal is corrupted or not by checking the syndrome $s(y) = Hy_b$, where $y_b = \text{bin}(\text{sign}(y))$ is the demodulated signal of $y$. Here, $\text{sign}(a)$ represents $+1$ if $a \geq 0$ and $-1$ otherwise and $\text{bin}(-1) = 1$, $\text{bin}(+1) = 0$. If $s(y)$ is a non-zero vector, it is detected that $y$ is corrupted during the transmission, and the decoder initiates the error correction process.

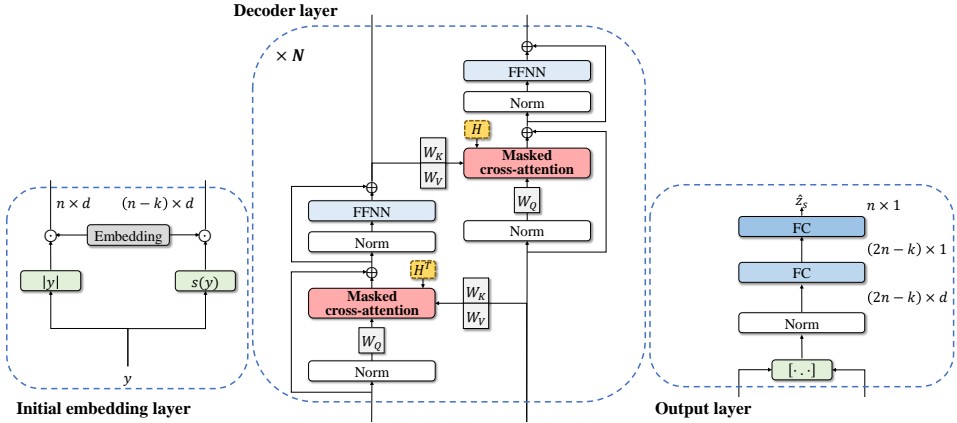

Figure 2: Architecture of CrossMPT.

## 3.2 ERROR CORRECTION CODE TRANSFORMER

ECCT is the first approach to present a model-free decoder with the transformer architecture. ECCT outperforms other neural decoders by employing the masked self-attention mechanism, whose mask matrix is determined by the code's PCM (Choukroun & Wolf, 2022a). A primary challenge in training transformer-based decoders is the issue of overfitting. In (Bennatan et al., 2018), the overfitting issue in model-free neural decoders is described as poor generalization to untrained codewords due to the exponentially large number of possible codewords. However, it has been resolved by a pre-processing technique that facilitates a syndrome-based decoding (Bennatan et al., 2018). It has been theoretically proven that, with this preprocessing step, the decoder's performance remains invariant to the specific codewords used in the training set (Bennatan et al., 2018).

As in (Bennatan et al., 2018), the preprocessing step of ECCT utilizes the magnitude and syndrome vectors to train multiplicative noise $\tilde{z}_s$, which is defined by

$$y = x_s + z = x_s\tilde{z}_s. \tag{1}$$

ECCT aims to estimate the multiplicative noise in (1), i.e., $f(y) = \hat{z}_s$. Then, the estimation of $x$ is $\hat{x} = \text{bin}(\text{sign}(yf(y)))$. If the multiplicative noise is correctly estimated such that $\text{sign}(\tilde{z}_s) = \text{sign}(\hat{z}_s)$, then $\hat{x}$ can be computed as:

$$\hat{x} = \text{bin}(\text{sign}(yf(y))) = \text{bin}(\text{sign}(x_s\tilde{z}_s\hat{z}_s)) = \text{bin}(\text{sign}(x_s)) = x.$$

ECCT employs a masked self-attention module to train the transformer architecture, where the input embedding is constructed by concatenating the magnitude and syndrome embeddings. As shown in Figure 1, the mask matrices of ECCT should clearly distinguish between necessary (unmasked) and unnecessary (masked) pairwise relationships among magnitude-magnitude, magnitude-syndrome, and syndrome-syndrome bit relations. In ECCT, the syndrome-syndrome part was only unmasked for self-relations, while the magnitude-syndrome part was unmasked based on the connections defined by the PCM. The magnitude-magnitude part, however, was unmasked for bit pairs connected at depth 2 (see Algorithm 1 in Choukroun & Wolf (2022a)). While the masking of magnitude-syndrome relations is intuitive, as it directly uses PCM, determining the relationships among magnitude themselves is not directly derivable from the PCM. Therefore, the algorithm for masking magnitude-magnitude part is neither straightforward nor unique. In Figure 1, the white areas indicate unmasked positions that require attention calculations, whereas the blue areas represent masked positions where attention calculations can be omitted. As the proportion of blue increases, the attention matrix becomes sparser, enhancing computational efficiency.

## 4 CROSS-ATTENTION MESSAGE-PASSING TRANSFORMER

In this section, we present the operational mechanism and architecture of CrossMPT. CrossMPT processes the magnitude and syndrome embeddings separately, applying a cross-attention mechanism to effectively capture their distinct information. It shares core principles with message passing

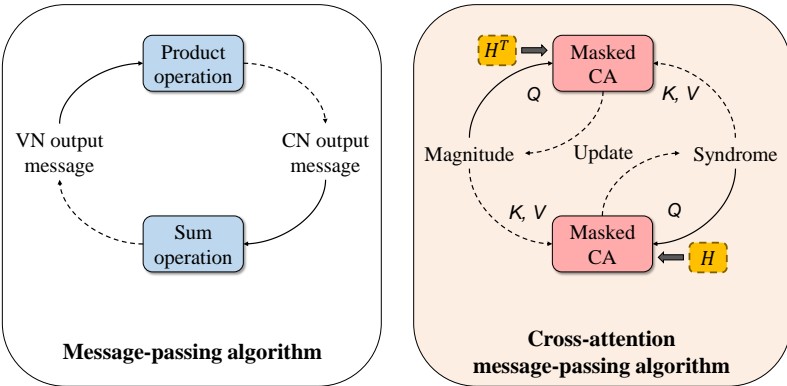

Figure 3: Conceptual comparison of the sum-product message-passing algorithm and the proposed cross-attention (CA) message-passing algorithm.

algorithm for decoding linear codes, iteratively updating the magnitude and syndrome embeddings of the received codewords. The overall architecture is depicted in Figure 2.

## 4.1 CROSS-ATTENTION MESSAGE-PASSING TRANSFORMER

One cross-attention block updates the magnitude embedding by using it as the query while generating the key and value from the syndrome embedding. Given this configuration, the attention map has the size $n \times (n - k)$, effectively representing the 'magnitude-syndrome' relation. To reflect this relationship, we employ the transpose of the PCM $H^\top$ as the mask matrix. This is because the $n$ rows of $H^\top$ correspond to the $n$ bit positions, and its $n - k$ columns are associated with the parity check equations, directly linking to $|y|$ and $s(y)$, respectively. The other cross-attention block similarly uses the syndrome embedding for the query, while the magnitude embedding generates the key and value. For this operation, we utilize the PCM $H$ as the mask matrix.

This configuration of separately processing two distinct informational properties resembles the message-passing decoding algorithm for decoding linear codes. Message-passing algorithms such as the sum-product algorithm (Richardson & Urbanke, 2001) are widely used for decoding LDPC codes. The message-passing algorithm operates by exchanging messages between variable nodes (VNs) and check nodes (CNs) over a Tanner (bipartite) graph (Richardson & Urbanke, 2001). In the Tanner graph, VNs convey information about the reliability of the received codeword, while CNs indicate the parity check equations. The edges between VNs and CNs represent the connections (relationships) between them. The message-passing decoder operates by exchanging messages between VNs and CNs via these edges. The output messages of VNs and CNs are updated in an iterative manner.

Similar to the principles of message-passing algorithms, CrossMPT iteratively updates the magnitude and syndrome embeddings. First, the magnitude embedding is updated using the masked cross-attention block, where the magnitude embedding for the query and syndrome embedding for the key and value. The syndrome embedding is updated in the subsequent masked cross-attention block, utilizing the previously updated magnitude embedding. In this block, the syndrome embedding is used for the query, while the updated magnitude embedding generates the key and value. The resulting output from this cross-attention block is the updated syndrome embedding. CrossMPT iteratively updates both the magnitude and syndrome embeddings to estimate the multiplicative noise accurately.

As a representative of the message-passing algorithm, Figure 3 depicts the sum-product algorithm and the cross-attention message-passing algorithm. In the sum-product algorithm, the VN output and CN output messages are iteratively updated using the sum and product operations. Similar to the sum-product algorithm, the magnitude and syndrome embeddings are iteratively updated using the masked cross-attention (Masked CA in the figure) blocks in CrossMPT. Note that $H$ and $H^\top$ serve as the mask matrices for these cross-attention blocks, while $Q$, $K$, and $V$ denote the query, key, and value of the cross-attention mechanism.

## 4.2 Model Architecture

In the initial embedding layer, we generate $|y| = (|y_1|, \ldots, |y_n|)$ and $s(y) = (s(y)_1, \ldots, s(y)_{n-k})$ from the received codeword, and project each element $y_i$ and $s(y)_i$ into $d$ dimension embedding row vectors $M_i$ and $S_i$, respectively, as follows:

$$M_i = |y_i| W_i, \qquad \text{for } i = 1, \ldots, n,$$
$$S_i = s(y)_i W_{i+n}, \quad \text{for } i = 1, \ldots, n-k,$$

where $W_i \in \mathbb{R}^{1 \times d}$ for $i = 1, \ldots, 2n - k$ denotes the trainable positional encoding vector.

These magnitude and syndrome embeddings are processed as separate inputs in the subsequent $N$ decoding layers. Each decoding layer contains two cross-attention blocks, each consisting of a cross-attention module, a feed-forward neural network (FFNN), and a normalization layer.

In the first cross-attention module, the attention module updates the 'magnitude' embedding by using the syndrome embedding. The query $Q_1$, key $K_1$, and value $V_1$ are assigned as follows:

$$Q_1 = MW_Q, K_1 = SW_K, V_1 = SW_V,$$

where $M = [M_1; \cdots; M_n] \in \mathbb{R}^{n \times d}$ and $S = [S_1; \cdots; S_{n-k}] \in \mathbb{R}^{(n-k) \times d}$ denote the magnitude and syndrome embeddings, respectively. Here, $W_Q, W_K, W_V$ denote the weight matrices for the query, key, and value, respectively. This architecture is referred to as the *cross-attention* message-passing transformer since the query corresponds to the magnitude embedding, while the key and value correspond to the syndrome embedding. Then, we employ the following scaled dot-product attention:

$$\text{Attention}(Q_1, K_1, V_1) = \text{softmax}\left( \frac{Q_1 K_1^\top + g(H^\top)}{\sqrt{d}} \right) V_1,$$

where $g(H^\top)$ is the mask matrix, and the function $g$ is defined as

$$g(A)_{i,j} = \begin{cases} 0 & \text{if } A_{i,j} = 1, \\ -\infty & \text{if } A_{i,j} = 0. \end{cases} \tag{2}$$

This configuration results in an attention map of size $n \times (n - k)$, representing the 'magnitude-syndrome' relationship. Therefore, we use the transpose of the PCM $H^\top$ as the mask matrix. Here, the $n$ rows of $H^\top$ correspond to the $n$ bit positions and the $n - k$ columns of $H^\top$ to the parity check equations, which are closely related to $|y|$ and $s(y)$, respectively. Finally, the output of this cross-attention module yields the updated magnitude embedding $M'$.

In the second cross-attention module, we update the 'syndrome' embedding using the updated magnitude embedding $M'$. In other words, the syndrome embedding serves as the query input, while $M'$ is used for both the key and value inputs. We use the *shared weight matrices* $W_Q, W_K, W_V$ from the first cross-attention module, and query $Q_2$, key $K_2$, and value $V_2$ are defined as follows:

$$Q_2 = SW_Q, K_2 = M'W_K, V_2 = M'W_V.$$

Here, the syndrome and magnitude embeddings correspond the rows and columns of the attention map, respectively. Thus, we employ the mask matrix $g(H)$, whose masking positions are zeros in $H$. Then, we apply the scaled dot-product attention and the resulting output provides the updated syndrome embedding $S'$. This updated syndrome embedding is utilized to further refine the magnitude embedding in the subsequent decoder layer, and this process is iteratively repeated across the $N$ decoder layers.

Finally, the updated magnitude and syndrome embeddings from the last decoder layer are concatenated and passed through a normalization layer and two fully connected (FC) layers. The first FC layer reduces the $(2n - k) \times d$ dimension embedding to a one-dimensional $2n - k$ vector, and the second FC layer further reduces the dimension from $2n - k$ into $n$. The final output provides an estimation of $\tilde{z}_s$. Since two cross-attention blocks of CrossMPT share the same weight matrices $W_Q, W_K, W_V$ and all other layers, CrossMPT has the same number of parameters as the original ECCT.

The objective of the proposed decoder is to learn the multiplicative noise $\tilde{z}_s$ in (1) and reconstruct the original transmitted signal $x$. We can obtain the multiplicative noise by $\tilde{z}_s = \tilde{z}_s x_s^2 = y x_s$. Then, the

Table 1: Comparison of decoding performance at three different $E_b/N_0$ (4 dB, 5 dB, 6 dB) for BP decoder, Hyper BP decoder (Nachmani & Wolf, 2019), AR BP decoder (Nachmani & Wolf, 2021), ECCT (Choukroun & Wolf, 2022a), and the proposed CrossMPT. The results are measured by the negative natural logarithm of BER. The best results are highlighted in **bold**. Higher is better.

| Architecture | | BP-based decoders | | | | | | | | | Model-free decoders | | | | | |
|---|---|---|---|---|---|---|---|---|---|---|---|---|---|---|---|---|
| | | BP | | | Hyp BP | | | AR BP | | | ECCT | | | CrossMPT | | |
| Codes | Parameter | 4 | 5 | 6 | 4 | 5 | 6 | 4 | 5 | 6 | 4 | 5 | 6 | 4 | 5 | 6 |
| BCH | (31,16) | 4.63 | 5.88 | 7.60 | 5.05 | 6.64 | 8.80 | 5.48 | 7.37 | 9.60 | 6.39 | 8.29 | 10.66 | **6.98** | **9.25** | **12.48** |
| | (63,36) | 4.03 | 5.42 | 7.26 | 4.29 | 5.91 | 8.01 | 4.57 | 6.39 | 8.92 | 4.86 | 6.65 | 9.10 | **5.03** | **6.91** | **9.37** |
| | (63,45) | 4.36 | 5.55 | 7.26 | 4.64 | 6.27 | 8.51 | 4.97 | 6.90 | 9.41 | 5.60 | 7.79 | 10.93 | **5.90** | **8.20** | **11.62** |
| | (63,51) | 4.5 | 5.82 | 7.42 | 4.8 | 6.44 | 8.58 | 5.17 | 7.16 | 9.53 | 5.66 | 7.89 | 11.01 | **5.78** | **8.08** | **11.41** |
| Polar | (64,32) | 4.26 | 5.38 | 6.50 | 4.59 | 6.10 | 7.69 | 5.57 | 7.43 | 9.82 | 6.99 | 9.44 | 12.32 | **7.50** | **9.97** | **13.31** |
| | (64,48) | 4.74 | 5.94 | 7.42 | 4.92 | 6.44 | 8.39 | 5.41 | 7.19 | 9.30 | 6.36 | 8.46 | 11.09 | **6.51** | **8.70** | **11.31** |
| | (128,64) | 4.1 | 5.11 | 6.15 | 4.52 | 6.12 | 8.25 | 4.84 | 6.78 | 9.3 | 5.92 | 8.64 | 12.18 | **7.52** | **11.21** | **14.76** |
| | (128,86) | 4.49 | 5.65 | 6.97 | 4.95 | 6.84 | 9.28 | 5.39 | 7.37 | 10.13 | 6.31 | 9.01 | 12.45 | **7.51** | **10.83** | **15.24** |
| | (128,96) | 4.61 | 5.79 | 7.08 | 4.94 | 6.76 | 9.09 | 5.27 | 7.44 | 10.2 | 6.31 | 9.12 | 12.47 | **7.15** | **10.15** | **13.13** |
| LDPC | (49,24) | 6.23 | 8.19 | 11.72 | 6.23 | 8.54 | 11.95 | 6.58 | 9.39 | 12.39 | 6.13 | 8.71 | 12.10 | **6.68** | **9.52** | **13.19** |
| | (121,60) | 4.82 | 7.21 | 10.87 | 5.22 | 8.29 | 13.00 | 5.22 | 8.31 | 13.07 | 5.17 | 8.31 | 13.30 | **5.74** | **9.26** | **14.78** |
| | (121,70) | 5.88 | 8.76 | 13.04 | 6.39 | 9.81 | 14.04 | 6.45 | 10.01 | 14.77 | 6.40 | 10.21 | 16.11 | **7.06** | **11.39** | **17.52** |
| | (121,80) | 6.66 | 9.82 | 13.98 | 6.95 | 10.68 | 15.80 | 7.22 | 11.03 | 15.90 | 7.41 | 11.51 | 16.44 | **7.99** | **12.75** | **18.15** |
| MacKay | (96,48) | 6.84 | 9.40 | 12.57 | 7.19 | 10.02 | 13.16 | 7.43 | 10.65 | 14.65 | 7.38 | 10.72 | 14.83 | **7.97** | **11.77** | **15.52** |
| CCSDS | (128,64) | 6.55 | 9.65 | 13.78 | 6.99 | 10.57 | 15.27 | 7.25 | 10.99 | 16.36 | 6.88 | 10.90 | 15.90 | **7.68** | **11.88** | **17.50** |
| Turbo | (132,40) | N/A | N/A | N/A | N/A | N/A | N/A | N/A | N/A | N/A | 4.74 | 6.54 | 9.06 | **5.55** | **7.92** | **10.94** |

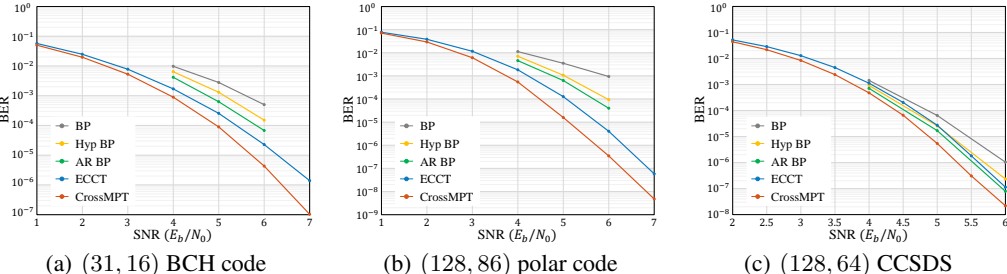

(a) $(31, 16)$ BCH code      (b) $(128, 86)$ polar code      (c) $(128, 64)$ CCSDS

Figure 4: The BER performance of various decoders (BP, Hyp BP, AR BP, ECCT) and CrossMPT.

target multiplicative noise for binary cross-entropy loss function is defined by $\tilde{z} = \text{bin}(\text{sign}(yx_s))$. Finally, the cross-entropy loss function for a received codeword $y$ is defined by

$$\mathcal{L} = -\sum_{i=1}^{n}\{\tilde{z}_i \log(1 - \sigma(f(y)_i)) + (1 - \tilde{z}_i)\log(\sigma(f(y)_i))\}.$$

To ensure a fair comparison between CrossMPT and ECCT, we adopt the same training setup used in the previous work (Choukroun & Wolf, 2022a). We use the Adam optimizer (Kingma & Ba, 2015) and conduct 1000 epochs. Each epoch consists of 1000 minibatches, where each minibatch is composed of 128 samples. All simulations were conducted using NVIDIA GeForce RTX 3090 GPU and AMD Ryzen 9 5950X 16-Core Processor CPU. The training sample $y$ is generated by $y = x_s + z$, where $x_s$ is the modulated signal corresponding to the all-zero codeword, and $z$ represents the AWGN channel noise, sampled from an normalized signal-to-noise ratio ($E_b/N_0$) range of 3 dB to 7 dB. The learning rate is initially set to $10^{-4}$ and gradually reduced to $5 \times 10^{-7}$ following a cosine decay scheduler.

## 5 EXPERIMENTAL RESULTS

In this section, we compare the proposed CrossMPT with the original ECCT across various code classes. Our experimental results do not include a comparison with the works of (Choukroun & Wolf, 2024a;b), as they have different objectives, such as generalizing the decoder to unseen codes (Choukroun & Wolf, 2024a) or jointly training the encoder and decoder (Choukroun & Wolf, 2024b). It is worth mentioning that our cross-attention architecture and the schemes of (Choukroun & Wolf,

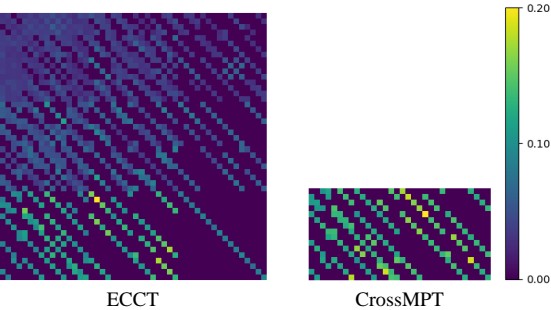

Figure 5: The average attention scores of all $N = 6$ layers for ECCT and CrossMPT.

2024a;b) are orthogonal methods, and combining them could present a promising direction for future research.

To verify the efficacy of CrossMPT, we train it for BCH codes, polar codes, turbo codes, and LDPC codes (including MacKay and CCSDS codes) and evaluate the bit error rate (BER) performance. All PCMs are taken from (Helmling et al., 2019). The implementation of the original ECCT is obtained from (Choukroun & Wolf, 2022b). For the testing, we collect at least 500 frame errors at each $E_b/N_0$ with random codewords. Table 1 compares the decoding performance of CrossMPT with the BP decoder, BP-based neural decoders (Nachmani & Wolf, 2019; 2021), and ECCT (Choukroun & Wolf, 2022a). The results of the BP-based decoders in Table 1 are obtained for 50 iterations. The results for both the proposed CrossMPT and ECCT, which are model-free decoders, are obtained with $N = 6$ and $d = 128$. For all types of codes, CrossMPT outperforms the conventional ECCT and all the other BP-based neural decoders. This improvement of CrossMPT is particularly notable in the case of LDPC codes. To provide more visual information, we plot the BER graphs for several codes in Figure 4.

An important aspect of our research is CrossMPT's capability to decode long codes (Appendix A), which remain beyond the reach of ECCT due to its high memory requirements, resulting from large attention maps. These results demonstrate the practical significance and architectural advantages of CrossMPT, proving its value in scenarios where ECCT encounters limitations. Especially, it achieves superior decoding performance for LDPC codes, outperforming the BP decoder with the maximum iteration of 100 (provided in Appendix B). Additionally, for short codes, CrossMPT closely approaches the optimal maximum likelihood (ML) decoding performance (provided in Appendix C). Additional experimental results for block error rate (BLER), comparison with successive cancellation list polar decoder, denoising diffusion ECCT (DDECCT), and the decoding performance for the Rayleigh channel, are provided in Appendices D, E, F, and G, respectively.

## 6 ABLATION STUDIES AND ANALYSIS

### 6.1 ANALYSIS OF ATTENTION MECHANISMS IN ECCT AND CROSSMPT

We provide an analysis of the attention scores in ECCT and CrossMPT. Figure 5 shows the average attention scores across $N = 6$ layers for both ECCT and CrossMPT on the $(32, 16)$ LDPC code (Abu-Surra et al., 2010). As shown in Figure 5, the attention map of ECCT reveals different importance among the relationships: Magnitude-magnitude, syndrome-syndrome, and magnitude-syndrome. One key observation is that the magnitude-magnitude and syndrome-syndrome relations exhibit relatively low attention scores compared to the magnitude-syndrome relation, which suggests that the magnitude-syndrome relationship is more significant than the others. Appendix H, in which we mask the magnitude-magnitude and syndrome-syndrome relationships, reveal no significant performance difference compared to when these relationships are not masked. This demonstrates that the conventional ECCT could be enhanced by focusing on the more critical relationships, as CrossMPT achieves this by eliminating the two relations with low attention scores and concentrating solely on the magnitude-syndrome relation. Therefore, we can claim that CrossMPT more efficiently focuses on the crucial aspect (i.e., magnitude-syndrome relation) compared to ECCT.

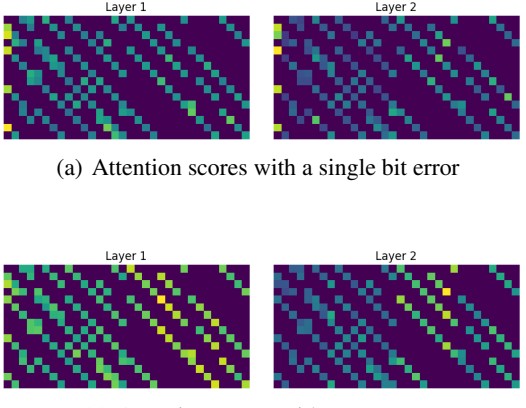

(a) Attention scores with a single bit error

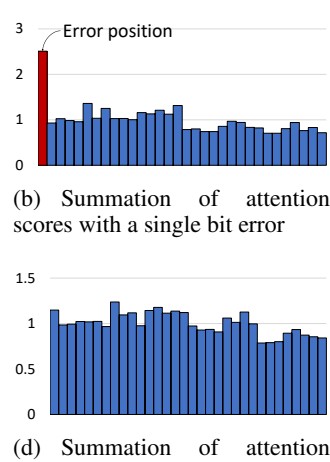

(b) Summation of attention scores with a single bit error

(c) Attention scores without an error

(d) Summation of attention scores without an error

Figure 6: The attention scores (a), (c) with a single bit error in the first bit position and without an error. The attention scores (b), (d) is carried out in the vertical direction.

## 6.2 VISUALIZATION OF CROSS-ATTENTION MAP

To further examine how CrossMPT operates, we intentionally corrupt a pre-determined bit of the $(32, 16)$ LDPC code and analyze the resulting attention maps. Figure 6 shows the attention scores for the first two layers and the summation of their attention scores when the *first bit* is corrupted. The summation is carried out vertically to demonstrate the attention score for each bit. As shown in Figure 6(b), the attention score of the first bit (or first column) is relatively higher than the others. However, once the error is corrected, CrossMPT no longer assigns high attention scores to that position (see Figure 12 in Appendix I). Figures 6(c) and 6(d) depict the attention scores and the summation of attention scores when no errors are present. Compared to the previous case, the attention scores are more uniformly distributed across all bit positions.

## 6.3 COMPLEXITY ANALYSIS

Two cross-attention blocks of CrossMPT share the same parameters for all decoder layers. They use the same weight matrices $W_Q, W_K, W_V$ for two cross-attention modules since the performance remains nearly identical even when the parameters are trained separately. Also, they share the parameters for the normalization layers and the FFNN layers. Thus, CrossMPT maintains the same number of parameters as the original ECCT.

Figure 1 illustrates the mask matrices of ECCT and CrossMPT. In the original ECCT, Figure 1(a) shows that a significant portion of the upper $n \times n$ submatrix is depicted in white, indicating that the most positions are unmasked. This $n \times n$ submatrix represents depth-2 connections in the Tanner graph (Choukroun & Wolf, 2022a), which results in an increase in the number of unmasked positions, thereby leading to a higher computational required. On the other hand, the lower $(n - k) \times n$ submatrix and the right $(n - k) \times n$ submatrix, which serve as the masking matrices for CrossMPT, are predominantly shown in blue, indicating that their attention matrices are sparser. Figure 7 compares the mask matrix density of CrossMPT and ECCT. For all codes, the mask matrix of CrossMPT is sparser than ECCT, which implies that CrossMPT can achieve lower computational complexity compared to the original ECCT.

The complexity of the self-attention mechanism of ECCT, without considering masking is, $\mathcal{O}(N(d^2(2n - k) + (2n - k)^2 d))$. When masking is taken into account, the complexity can be reduced to $\mathcal{O}(N(d^2(2n - k) + hd))$ (Choukroun & Wolf, 2022a), where $h = \rho_1(2n - k)^2$ denotes the fixed number of computations of the self-attention module and $\rho_1$ denotes the density of the mask matrix in ECCT. Similarly, the complexity of the two cross-attention modules of CrossMPT, without considering the masking, is $\mathcal{O}(N(d^2(2n - k) + 2n(n - k)d))$. When masking is taken into account, the complexity can be reduced to $\mathcal{O}(N(d^2(2n - k) + (2\tilde{h})d))$, where $\tilde{h} = \rho_2 n(n - k)$ denotes the number of computations of a single cross-attention module and $\rho_2$ denotes the density

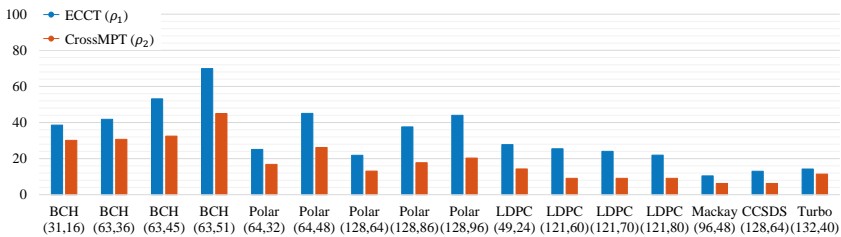

Figure 7: Comparison of the mask matrix density between ECCT and CrossMPT.

Table 2: Comparison of FLOPs, inference time, and training time between ECCT and CrossMPT for various codes. Inference time is measured for decoding a single codeword and training time is measured for a single epoch.

| Codes | Parameter | FLOPs | | Inference (codeword) | | Training (epoch) | | Mask density | | Memory usage | |
|---|---|---|---|---|---|---|---|---|---|---|---|
| | | CrossMPT | ECCT | CrossMPT | ECCT | CrossMPT | ECCT | CrossMPT | ECCT | CrossMPT | ECCT |
| BCH | (63,45) | 99.8 M | 106.4 M | 326 $\mu$s | 328 $\mu$s | 29 s | 29 s | 32.45% | 53.09% | 962 MiB | 1828 MiB |
| LDPC | (121,70) | 229.7 M | 256.8 M | 400 $\mu$s | 450 $\mu$s | 58 s | 80 s | 9.09% | 24.01% | 1980 MiB | 3926 MiB |
| | (121,80) | 212.5 M | 238.0 M | 391 $\mu$s | 436 $\mu$s | 53 s | 76 s | 9.09% | 21.94% | 1936 MiB | 3602 MiB |
| Turbo | (132,40) | 303.6 M | 343.4 M | 459 $\mu$s | 511 $\mu$s | 83 s | 110 s | 11.43% | 14.25% | 2362 MiB | 5580 MiB |
| BCH | (255,223) | 28.2 M | 53.5 M | 747 $\mu$s | 859 $\mu$s | 56 s | 145 s | 48.63% | 78.21% | 1036 MiB | 7318 MiB |
| WRAN | (384,320) | 53.1 M | 111.3 M | 1295 $\mu$s | 1638 $\mu$s | 104 s | 305 s | 5.21% | 13.25% | 3270 MiB | 18192 MiB |

of the mask matrix in CrossMPT. Furthermore, since $\rho_1 > \rho_2$ as shown in Figure 7, we conclude that $h > 2\tilde{h}$, which indicates that CrossMPT achieves a reduction in computational complexity compared the original ECCT.

Table 2 compares the total FLOPs, inference time, training time, and memory usage between ECCT and CrossMPT. The inference time refers to the duration required to decode a single codeword and the training time measures the duration to complete one epoch of training. All results are obtained for $N = 6$ and $d = 128$, except for (255,223) BCH code and (384,320) WRAN LDPC code, which are obtained for $N = 6$ and $d = 32$. For all three metrics, CrossMPT outperforms ECCT. Since inference and training times are closely related to the FLOPs, a reduction in FLOPs directly leads to shorter inference and training times. Notably, CrossMPT significantly reduces memory usage compared to ECCT, especially for long codes. This improvement arises from the reduced size of the attention map; $2n(n - k)$ for CrossMPT and $(2n - k)^2$ for ECCT. The results in Tables 1 and 2 demonstrate that the proposed CrossMPT not only improves the decoding performance but also significantly reduces FLOPs, inference time, training time, and memory usage compared to the original ECCT. Additional analysis of the training time required to achieve the target loss is provided in Appendix J.

CrossMPT's sequential decoding approach may limit its throughput in certain scenarios. However, pipelining (Li et al., 2021b; Rowshan et al., 2024) enables CrossMPT to effectively increase its throughput when decoding multiple codewords (see Appendix K).

## 7 CONCLUSION

We developed a novel transformer architecture for ECC decoding called CrossMPT, which improves both decoding performance and computational efficiency. CrossMPT achieves this by adopting a more effective architecture that processes magnitude and syndrome embeddings through the cross-attention mechanism. This approach leverages the structured representation of codeword bit relationships in the PCM, enabling the model to accurately learn these relationships while significantly reducing memory usage, FLOPs, inference time, and training time. Most existing transformer-based decoders have been limited to short codes due to challenges in training long codes, primarily caused by high memory usage and computational complexity. However, CrossMPT effectively addresses these challenges, improving the viability of transformer-based decoders for long codes.

ACKNOWLEDGMENTS

This work was supported by Institute of Information & Communications Technology Planning & Evaluation (IITP) grant funded by the Korean Government (MSIT) (RS-2024-00398449, Network Research Center: Advanced Channel Coding and Channel Estimation Technologies for Wireless Communication Evolution) and the National Research Foundation of Korea (NRF) grant funded by the Korean Government (MSIT) (No. RS-2023-00212103).

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

# A  PERFORMANCE ON LONGER CODES

We present the BER performance for three longer codes in Figures 8(a), 8(b), and 8(c) for ECCT and CrossMPT $N = 6$, $d = 32$. For all three codes ((a) (529,440) LDPC code, (b) (384,320) wireless regional area network (WRAN) LDPC code, (c) (512,384) polar code), the proposed CrossMPT outperforms the original ECCT. Despite its reduced complexity, CrossMPT significantly enhances the decoding performance compared to ECCT, not only for short-length codes but also for longer codes. Also, Figures 8(d) and 8(e) show the decoding performance of CrossMPT for much longer codes. The BER performances of the (648,540) IEEE 802.11n LDPC code ($N = 10$, $d = 128$) and (1056,880) WiMAX LDPC code ($N = 6$, $d = 32$) demonstrate that CrossMPT efficiently trains how to decode the codeword even for large $N$ and $d$ and performs well for longer codes. Again, we emphasize CrossMPT's capability to decode long codes where ECCT struggles due to high memory allocation (large attention map). The structure of CrossMPT demonstrates its efficiency in learning long codes, surpassing the limitations of short or moderate codelengths of transformer-based decoders.

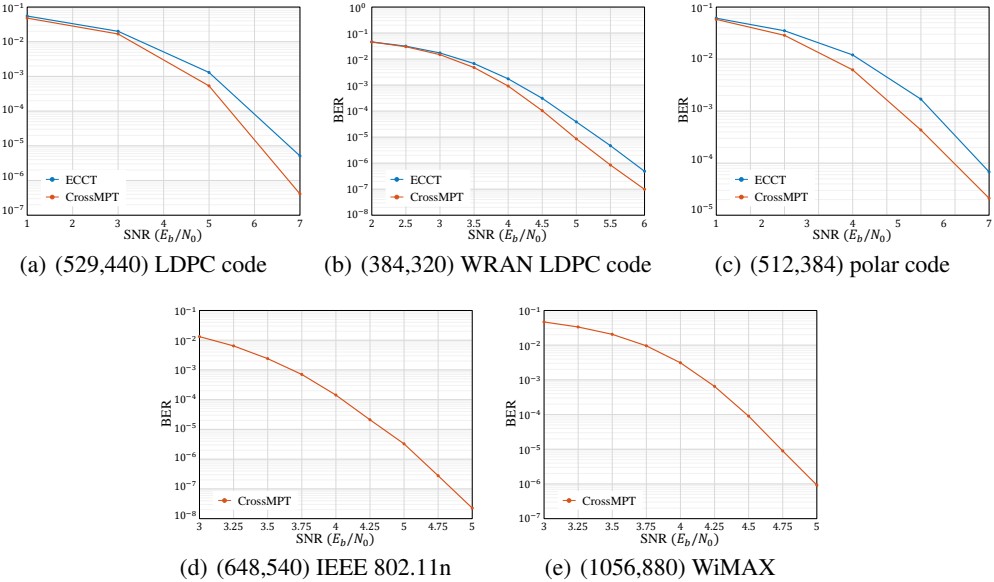

Figure 8: The decoding performance of long codes.

## B  COMPARISON WITH THE BP DECODER

Figure 9 shows the decoding performance between the traditional BP decoder with a maximum number of iterations of 20, 50, and 100 and CrossMPT for both short and long LDPC codes. Figures 9(a) and 9(b) compare the BER performance for (121,80) LDPC codes ($N = 6$, $d = 128$) and (648,540) IEEE 802.11n LDPC code ($N = 10$, $d = 128$), respectively. Notably, the proposed CrossMPT can outperform the BP decoder for both short and long LDPC codes. These results highlight that CrossMPT efficiently trains how to decode the codeword across a wide range of code lengths.

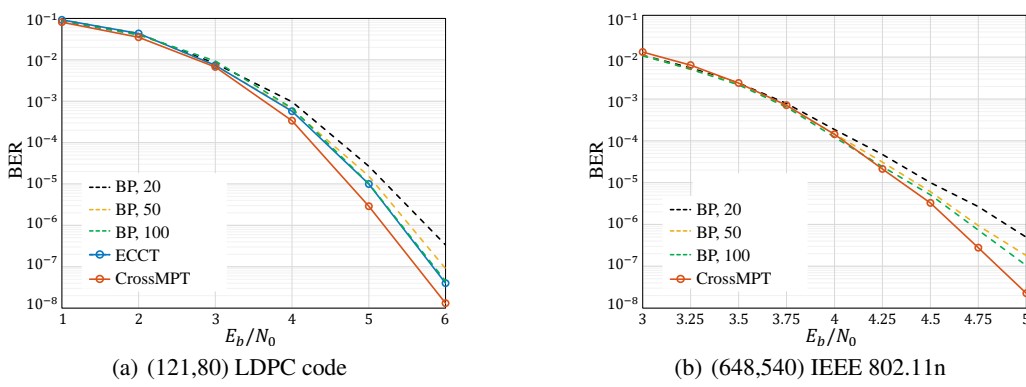

(a) (121,80) LDPC code          (b) (648,540) IEEE 802.11n

Figure 9: Performance comparison between BP decoder (iteration 20, 50, and 100) and CrossMPT.

## C  COMPARISON WITH THE ML DECODER

We compare ECCT and CrossMPT with the ML decoder for short BCH codes. Figure 10 demonstrates the BER performance of $(31, 16)$ BCH code and $(31, 21)$ BCH code. Especially, these results show that CrossMPT closely approaches the optimal ML performance for short codes.

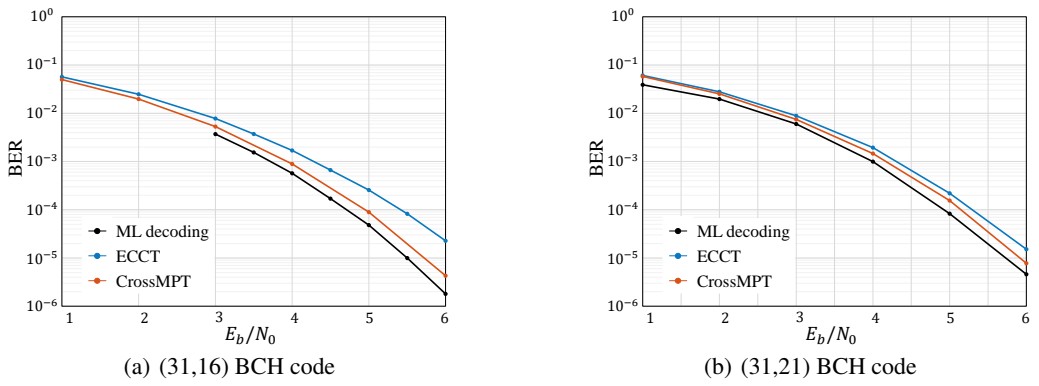

(a) (31,16) BCH code          (b) (31,21) BCH code

Figure 10: The decoding performance comparison between ML decoder, ECCT, and CrossMPT.

## D  BLOCK ERROR RATE PERFORMANCE

Table 3 demonstrates BLER results for various code classes ($N = 6$, $d = 128$). Also, Figure 11 shows the BLER performance of (31,16) BCH code, (63,51) BCH code, and (648,540) IEEE 802.11n LDPC code. For BCH codes, we compare the decoding performance of CrossMPT with the traditional Berlekamp-Massey (BM) decoder, maximum likelihood (ML) decoding algorithm, ECCT. For LDPC codes, we compare the decoding performance of CrossMPT with the traditional

BP decoder with a maximum number of iterations of 20, 50, and 100. As shown in the table, CrossMPT outperforms ECCT in the BLER performance and also has a comparable BLER results compared to the traditional decoding algorithms.

In addition, the traditional decoders are code-specific decoders, tailored to each class of codes. For example, LDPC codes are effectively decoded by the BP decoder, BCH codes by the BM decoder, and polar codes by the SCL decoder. However, unlike the traditional decoders, a key advantage of CrossMPT is its versatility. While conventional decoders are good and valid only for respective code classes, CrossMPT performs effectively across a wide range of code classes. This universality highlights the broader applicability and potential of CrossMPT in various decoding scenarios and future communication paradigm such and semantic communication.

The complexity of transformer-based decoders is relatively high compared to code-specific decoders. Reducing their computational requirements will be an important focus for future work.

Table 3: The BLER results for ECCT and CrossMPT. The results are measured by the negative natural logarithm of BLER.

| Method | ECCT | | | CrossMPT | | |
|---|---|---|---|---|---|---|
| $E_b/N_0$ | 4 | 5 | 6 | 4 | 5 | 6 |
| (31,16) BCH | 4.19 | 5.98 | 8.16 | 5.12 | 7.24 | 10.31 |
| (63,36) BCH | 2.43 | 4.10 | 6.40 | 2.50 | 4.23 | 6.61 |
| (63,45) BCH | 2.75 | 4.75 | 7.67 | 3.18 | 5.33 | 8.66 |
| (63,51) BCH | 2.72 | 4.85 | 7.82 | 2.94 | 5.16 | 8.36 |
| (64,32) Polar | 4.18 | 6.47 | 9.07 | 4.83 | 7.26 | 10.41 |
| (64,48) Polar | 3.08 | 5.06 | 7.60 | 3.56 | 5.68 | 8.35 |
| (49,24) LDPC | 3.62 | 5.89 | 9.39 | 4.47 | 7.20 | 10.68 |
| (121,70) LDPC | 3.21 | 6.69 | 11.91 | 4.28 | 8.42 | 13.35 |

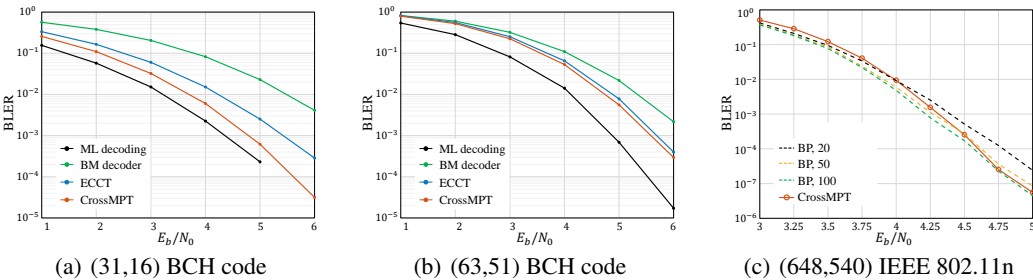

| (a) (31,16) BCH code | (b) (63,51) BCH code | (c) (648,540) IEEE 802.11n |
|---|---|---|

Figure 11: The BLER performance comparison between the traditional decoders and CrossMPT.

# E COMPARISON WITH SUCCESSIVE CANCELLATION LIST POLAR DECODER

We compare the BER performance of the SCL decoder, ECCT, and CrossMPT in Table 4. The performance of the SCL decoder is from (Choukroun & Wolf, 2022a). Although the contribution of $L$ is significant in long codes, the SCL decoder achieves a great performance with small $L$, such as $L = 4$. As reported in (Choukroun & Wolf, 2022a; 2023), the SCL decoder outperforms ECCT. This is because the SCL decoder is a decoder specialized for Polar codes and is a state-of-the-art algorithm that has undergone extensive development over a long period. CrossMPT has made significant improvements from ECCT and even outperforms the SCL decoder for (64,48) polar code.

# F COMPARISON WITH DDECCT

For a fair comparison with DDECCT, we also apply the denoising diffusion training technique to CrossMPT. Table 5 compares the BER performance of ECCT (Choukroun & Wolf, 2022a), CrossMPT, DDECCT, and CrossMPT applying the denoising diffusion model. All four decoders are model-free decoders using the transformer architecture, and simulations are taken for $N = 6$,

Table 4: Comparison of decoding performance at three different $E_b/N_0$ (4 dB, 5 dB, 6 dB) for SCL decoder, ECCT, and CrossMPT. The results are measured by the negative natural logarithm of BER. The best results are highlighted in **bold** and the second best is underlined. Higher is better.

| Method | SCL ($L=1$) | | | SCL ($L=4$) | | | ECCT | | | CrossMPT | | |
|---|---|---|---|---|---|---|---|---|---|---|---|---|
| Parameter | 4 | 5 | 6 | 4 | 5 | 6 | 4 | 5 | 6 | 4 | 5 | 6 |
| (64,32) | 7.30 | 9.67 | 13.18 | **8.11** | **10.70** | **14.04** | 6.99 | 9.44 | 12.32 | 7.50 | 9.97 | 13.31 |
| (64,48) | 6.19 | 8.41 | 10.97 | **6.69** | 8.63 | 11.24 | 6.36 | 8.46 | 11.09 | 6.51 | **8.70** | **11.31** |
| (128,64) | 8.37 | 11.69 | 13.70 | **9.60** | **13.16** | **17.42** | 5.92 | 8.64 | 12.18 | 7.52 | 11.21 | 14.76 |
| (128,86) | 7.54 | 10.74 | 15.14 | **9.26** | **13.04** | **17.13** | 6.31 | 9.01 | 12.45 | 7.86 | 11.45 | 15.47 |
| (128,96) | 6.74 | 9.53 | 13.53 | **8.02** | **11.60** | **18.16** | 6.31 | 9.12 | 12.47 | 7.15 | 10.15 | 13.13 |

$d = 128$. We conduct simulations for codes where DDECC performs better than CrossMPT. For the rest of the codes, CrossMPT outperforms DDECC. The proposed CrossMPT shows superior decoding performance compared to the original ECCT. Compared to DDECC, CrossMPT demonstrates similar BER performance for polar codes, but it even outperforms DDECC for BCH and LDPC codes. When the denoising diffusion technique is applied to CrossMPT, it achieves the best performance among others, where DDECC, CrossMPT, and ECCT follow. This proves that the CrossMPT architecture provides separate gain from the denoising diffusion algorithm for transformer-based decoders.

Table 5: Comparison of decoding performance at three different $E_b/N_0$ (4 dB, 5 dB, 6 dB) for ECCT (Choukroun & Wolf, 2022a), CrossMPT, and DDECC (Choukroun & Wolf, 2023). The results are measured by the negative natural logarithm of BER. The best results are highlighted in **bold** and the second best is underlined. Higher is better.

| Architecture | | *Without* denoising diffusion | | | | | | *With* denoising diffusion | | | | | |
|---|---|---|---|---|---|---|---|---|---|---|---|---|---|
| Codes | Parameter | ECCT | | | CrossMPT | | | ECCT | | | CrossMPT | | |
| | | 4 | 5 | 6 | 4 | 5 | 6 | 4 | 5 | 6 | 4 | 5 | 6 |
| BCH | (63,36) | 4.86 | 6.65 | 9.10 | 5.03 | 6.91 | 9.37 | 5.11 | 7.09 | 9.82 | **5.23** | **7.20** | **10.01** |
| Polar | (128,64) | 5.92 | 8.64 | 12.18 | 7.52 | 11.21 | 14.76 | 9.11 | 12.9 | 16.30 | **10.21** | **13.63** | **17.28** |
| | (128,86) | 6.31 | 9.01 | 12.45 | 7.51 | 10.83 | 15.24 | 7.60 | 10.81 | 15.17 | **8.56** | **12.04** | **15.37** |
| | (128,96) | 6.31 | 9.12 | 12.47 | 7.15 | 10.15 | 13.13 | 7.16 | 10.3 | 13.19 | 7.57 | 10.61 | **13.33** |
| MacKay | (96,48) | 7.38 | 10.72 | 14.83 | 7.97 | 11.77 | 15.52 | 8.12 | 11.88 | 15.93 | **8.85** | **12.58** | **17.69** |

# G DECODING PERFORMANCE FOR RAYLEIGH FADING CHANNEL

The original ECCT architecture shows robustness to non-Gaussian channels (e.g., Rayleigh fading channel) (Choukroun & Wolf, 2022a, Supplementary). We also measured the decoding performance of CrossMPT in Rayleigh fading channels. To compare with ECCT, we use the same fading channel as in (Choukroun & Wolf, 2022a). The received codeword is given as $y = hx + z$, where $h$ is an $n$-dimensional i.i.d. Rayleigh distributed vector with a scale parameter $\alpha = 1$ and $z \sim N(0, \sigma^2)$. The following table demonstrates the BER performance of ECCT and CrossMPT in Rayleigh fading channel and CrossMPT still outperforms the original ECCT architecture for all types of codes.

| Codes | (31,16) BCH | | | (64,32) Polar | | | (128,64) Polar | | | (128,86) Polar | | | (121,70) LDPC | | | (128,64) CCSDS | | |
|---|---|---|---|---|---|---|---|---|---|---|---|---|---|---|---|---|---|---|
| Methods | 4 | 5 | 6 | 4 | 5 | 6 | 4 | 5 | 6 | 4 | 5 | 6 | 4 | 5 | 6 | 4 | 5 | 6 |
| ECCT | 5.18 | 6.04 | 6.92 | 5.53 | 6.62 | 7.80 | 4.31 | 5.37 | 6.63 | 4.02 | 4.81 | 5.70 | 3.91 | 4.97 | 6.31 | 2.46 | 3.97 | 5.79 |
| CrossMPT | 5.53 | 6.55 | 7.61 | 5.91 | 7.17 | 8.48 | 4.70 | 5.93 | 7.34 | 4.41 | 5.38 | 6.46 | 4.25 | 5.53 | 7.11 | 5.25 | 6.94 | 8.92 |

## H   ABLATION STUDY WITH ADDITIONAL MASKING

To understand the impact of magnitude-magnitude and syndrome-syndrome relationships, we ex-ammine the decoding performance of ECCT with additional masking applied to all positions cor-responding to these relationships. Table 6 compares the decoding performance of ECCT with this additional masking, standard ECCT, and CrossMPT. The results show no significant performance degradation with the additional masking, indicating that the magnitude-magnitude and syndrome-syndrome relationships are not critical to decoding performance.

Table 6: Comparison of decoding performance at three different $E_b/N_0$ (4 dB, 5 dB, 6 dB) for ECCT with additional masking, standard ECCT, and CrossMPT. The results are measured by the negative natural logarithm of BER. The best results are highlighted in **bold**. Higher is better.

| Method | ECCT + Masking | | | ECCT | | | CrossMPT | | |
|---|---|---|---|---|---|---|---|---|---|
| Parameter | 4 | 5 | 6 | 4 | 5 | 6 | 4 | 5 | 6 |
| $(31, 16)$ BCH | 6.52 | 8.55 | 11.42 | 6.39 | 8.29 | 10.66 | **6.98** | **9.25** | **12.48** |
| $(63, 45)$ BCH | 5.53 | 7.74 | 10.88 | 5.60 | 7.79 | 10.93 | **5.90** | **8.20** | **11.62** |
| $(64, 48)$ Polar | 6.25 | 8.26 | 10.93 | 6.36 | 8.46 | 11.09 | **6.51** | **8.70** | **11.31** |
| $(121, 60)$ LDPC | 4.98 | 7.91 | 12.61 | 5.17 | 8.31 | 13.30 | **5.74** | **9.26** | **14.78** |

## I   VISUALIZATION OF CROSS-ATTENTION MAP

Figure 12 illustrates the attention scores for all $N = 6$ layers with a single bit error (bit error in the *first position*). The first three layers have relatively high attention score at the error position (first bit). Then, when the error is corrected, the attention score becomes lower at the last three layers.

For model-based neural decoders, interpretation and analysis in terms of graphs are feasible be-cause their structural architecture is inherently based on conventional graph-based decoding al-gorithms (Wang et al., 2021; Ankireddy & Kim, 2023). However, in the case of model-free ap-proaches, it remains challenging to determine how attention scores or weights are assigned to spe-cific nodes and how these assignments are influenced by graph properties such as node degree.

Although we analyzed how the attention scores change depending on where the error occurs in Figure 12, we have not yet achieved a rigorous analysis beyond this level. This remains a critical issue in model-free approaches and represents a problem that needs to be addressed in future work.

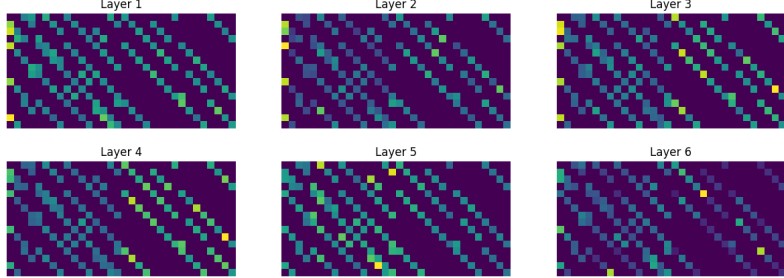

Figure 12: Attention scores of $N = 6$ layers with a single bit error.

## J   TRAINING TIME TO ACHIEVE THE TARGET LOSS

In Table 7, we compare the training time required for ECCT and CrossMPT to achieve the target loss. The target loss is set as a minimum loss of ECCT during the training. For $(128, 86)$ polar code, the minimum loss of ECCT during the 1000 epochs is $2.28 \times 10^{-2}$. To achieve the loss $2.28 \times 10^{-2}$, CrossMPT takes 6912 s, while ECCT requires 72917 s. Similarly, for $(128, 64)$ CCSDS code,

the minimum loss of ECCT is $2.79 \times 10^{-2}$ and CrossMPT requires 2356 s, while ECCT requires 85770 s. These results demonstrate that CrossMPT achieves the target loss significantly faster than ECCT, highlighting its efficiency in terms of training time.

Table 7: Comparison of training time to achieve the target loss for ECCT and CrossMPT.

| Codes | Methods | Target loss | Time |
|---|---|---|---|
| (128,86) Polar | ECCT | $2.28 \times 10^{-2}$ | 72917 s |
| | CrossMPT | $2.28 \times 10^{-2}$ | 6912 s |
| (128,64) CCSDS | ECCT | $2.79 \times 10^{-2}$ | 86770 s |
| | CrossMPT | $2.79 \times 10^{-2}$ | 2356 s |

## K    THROUGHPUT ANALYSIS

Table 2 demonstrates that CrossMPT outperforms ECCT in terms of inference time. However, while a fully parallel processor can accelerate ECCT, the sequential decoding architecture of CrossMPT limits its potential for throughput improvement. To address this, a pipelining approach–commonly employed in various ECC decoders (Li et al., 2021b; Rowshan et al., 2024)–can be applied to maximize CrossMPT's decoding throughput (see Figure 3 in (Li et al., 2021b)). By unrolling two cross-attention blocks, CrossMPT can simultaneously process two consecutive codewords across two cross-attention blocks within the same layer. This means that while the second cross-attention block processes the first codeword, the first cross-attention block concurrently decodes the subsequent codeword. This pipelining strategy ensures that CrossMPT maintains speed advantages over ECCT, in fully parallel scenarios. Figure 13 provides an example of decoding multiple codewords in CrossMPT with $N = 2$.

In wireless communications, the decoder's throughput is often a more critical concern than latency. This is because the latency from communication protocols and signal processing in preceding receiver blocks would be longer than the latency introduced by the channel decoder. Throughput becomes especially important when supporting very high data rates in wireless communication as the channel decoder can be a bottleneck.

Furthermore, in wireless communication scenarios, a sequential algorithm may be preferred for its enhanced performance or reduced complexity. The layered decoding algorithm for LDPC codes has been widely adopted as a de facto standard (Bae et al., 2019; Hailes et al., 2015; Li et al., 2021a), despite its sequential nature and limitation on parallelism, exemplifying the preference for sequential algorithms. The layered decoding algorithm is favored for its superior decoding performance compared to fully parallel sum-product decoding at equivalent computational complexities.

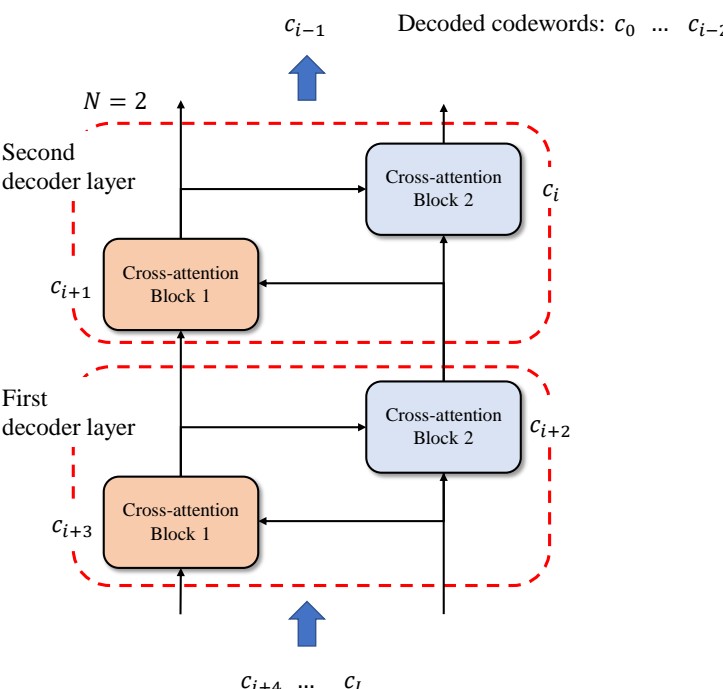

Figure 13: Example of decoding multiple codewords in CrossMPT with $N = 2$.

