# OpenReview forum: "CrossMPT: Cross-attention Message-passing Transformer for Error Correcting Codes"
_ICLR.cc/2025/Conference — ICLR 2025 Poster_

### Official Review · Reviewer_HjdF · 2024-10-27

**Soundness:** 3
**Presentation:** 3
**Contribution:** 2
**Rating:** 6
**Confidence:** 5

**Summary:**

The paper presents an extension of the existing ECCT by splitting, as with classical message-passing decoding (BP), the self-attention module between variable and check nodes. This is done similarly to classical Transformers decoders via the additional cross-attention while the original ECCT has only self-attention layers.

**Strengths:**

The paper is well-written and clear.

The paper is well-presented and has extensive ablation studies.

The method is interesting and sounds natural given classical message-passing decoders.

The method improves over the original ECCT and allows more memory efficient training on GPUs.

**Weaknesses:**

I'm concerned about the complexity analysis and thus the claim of superiority of the method over ECCT.

Cross-MPT adopts a **sequential** processing approach that appears **after** the self-attention, which is the only module in ECCT.
This sequential processing (1 self-attention + 1 cross-attention) is then necessarily slower than ECCT's single parallel processing (1 self-attention only) on modern parallel computing HW.

It obviously allows better decoding performance, making the comparison unfair, even if the parameters are shared.

**Questions:**

1 - The authors should integrate clearer information regarding the complexity as explained in the Weakness section.

2 - There is a large gap remaining with SCL decoding. The authors should explain and investigate why the (cross) ECCT and its variants struggle with it.

3 - The authors should provide comparisons between the methods for various dimensions/layers, as done with the ECCT and its extensions.

---

> ### Author Response · Authors · 2024-11-22
> **Response**
>
> ## Complexity analysis of CrossMPT
>
> First, CrossMPT consists of *two cross-attention layers* (not 1 self-attention + 1 cross-attention), which are significantly smaller in size compared to a single self-attention layer in ECCT: $2n(n-k)$ for two cross-attention layers and $(2n-k)^2$ for a single self-attentioin. Therefore, the overall size of two cross-attention modules is smaller than that of a single self-attention module in ECCT, which could explain the reduction in FLOPs, inference time, and training time, as shown in Table 2.
>
> As the reviewer pointed out, CrossMPT's sequential processing may differ from parallel processing. However, as mentioned in Figure 13, Appendix J, when decoding multiple codewords, the application of pipelining ensures that CrossMPT's decoding speed does not lag behind ECCT. As the reviewer mentioned, we would include these points in the revised manuscript.
>
> ## Comparison with SCL decoder
>
> Thank you for the comments for SCL decoding. Polar codes have been extensively studied over a long period, and the SCL decoder has become a mature and highly optimized decoding technique. It is highly effective for polar codes and achieves excellent performance.
> However, considering that transformer-based decoders are still in their early stages of research, their performance is quite promising, and there is a significant potential for improvement in both decoding performance and complexity.
>
> In addition, the traditional decoders are code-specific decoders, tailored to each class of codes. For example, LDPC codes are effectively decoded by the BP decoder, BCH codes by the BM decoder, and polar codes by the SCL decoder. A key advantage of ECCT and CrossMPT is its versatility. While conventional decoders are good and valid only for respective code classes, CrossMPT, as a transformer-based universal decoder, performs effectively across a wide range of code classes. This universality highlights the broader applicability and potential of CrossMPT in various decoding scenarios and future communication paradigm such and semantic communication.
>
> ## Decoding performance for various dimensions/layers
>
> We conducted additional simulations for $N={2,6}$ and $d={64,128}$ for (31,16) BCH code, (63,45) BCH code, (64,32) polar code, and (49,24) LDPC code. As shown in the table, CrossMPT with $d=64$ achieves better decoding performance than $d=128$ ECCT with the same number of decoder layer ($N$).
>
> | $N,d$ |  |  | $N=2$ | $d=64$ |  |  |  |  | $N=2$ | $d=128$ |  |  |  |  | $N=6$ | $d=64$ |  |  |  |  | $N=6$ | $d=128$ |  |  |
> | --- | --- | --- | --- | --- | --- | --- | --- | --- | --- | --- | --- | --- | --- | --- | --- | --- | --- | --- | --- | --- | --- | --- | --- | --- |
> | Method |  | ECCT |  |  | CrossMPT |  |  | ECCT |  |  | CrossMPT |  |  | ECCT |  |  | CrossMPT |  |  | ECCT |  |  | CrossMPT |  |
> | SNR | 4 dB | 5 dB | 6 dB | 4 dB | 5 dB | 6 dB | 4 dB | 5 dB | 6 dB | 4 dB | 5 dB | 6 dB | 4 dB | 5 dB | 6 dB | 4 dB | 5 dB | 6 dB | 4 dB | 5 dB | 6 dB | 4 dB | 5 dB | 6 dB |
> | (31,16) BCH | 4.78 | 5.28 | 7.01 | 5.33 | 6.91 | 9.07 | 5.18 | 6.82 | 8.91 | 5.72 | 7.60 | 9.69 | 5.85 | 7.52 | 10.08 | 6.65 | 8.85 | 11.83 | 6.39 | 8.29 | 10.66 | 6.98 | 9.25 | 12.48 |
> | (63,45) BCH | 4.66 | 6.16 | 8.17 | 4.97 | 6.78 | 9.33 | 4.79 | 6.39 | 8.49 | 5.11 | 6.96 | 9.60 | 5.41 | 7.49 | 10.25 | 5.63 | 7.83 | 10.93 | 5.60 | 7.79 | 10.93 | 5.90 | 8.20 | 11.62 |
> | (64,32) Polar | 4.57 | 5.86 | 7.50 | 5.20 | 6.75 | 8.68 | 4.87 | 6.20 | 7.93 | 5.52 | 7.21 | 9.43 | 6.48 | 8.60 | 11.43 | 7.05 | 9.48 | 12.26 | 6.99 | 9.44 | 12.32 | 7.50 | 9.97 | 13.31 |
> | (49,24) LDPC | 4.58 | 6.18 | 8.46 | 5.14 | 7.08 | 10.05 | 4.71 | 6.38 | 8.73 | 5.32 | 7.38 | 10.15 | 5.91 | 8.42 | 11.90 | 6.42 | 9.08 | 12.81 | 6.13 | 8.71 | 12.10 | 6.68 | 9.52 | 13.19 |

---

> > ### Comment · Reviewer_HjdF · 2024-11-23
> >
> > I thank the authors for their response and rebuttal.
> >
> > Maybe I am missing it but I cannot see in the revised manuscript where the authors admit the sequential processing hinders the complexity analysis (or any other limitation analysis).
> > Also, there is no reference to Appendix J in the text.

---

> ### Author Response · Authors · 2024-11-26
> **Response**
>
> Thank you for the comments, which have helped us improve our manuscript.
>
> We have revised the manuscript accordingly and uploaded the updated version to OpenReview. Additionally, we noted that the previous version of our manuscript mentioned Appendix J as "Additional analysis on training convergence and throughput of CrossMPT is provided in Appendix I and J, respectively." To provide a more detailed explanation and improve clarity, we have incorporated a new paragraph about sequential processing in the last paragraph of "Complexity Analysis," which is as follows:
>
> >CrossMPT's sequential decoding approach may limit its throughput in certain scenarios.
> However, pipelining [1], [2] enables CrossMPT to effectively increase its throughput when decoding multiple codewords (see Appendix K).
> >
>
> We also added references [1], [2] and detailed explanation to the pipelining strategy employed in ECC decoders in **Appendix K** of the revised manuscript as follows:
>
> > Table 2 demonstrates that CrossMPT outperforms ECCT in terms of inference time. However, while a fully parallel processor can accelerate ECCT, the sequential decoding architecture of CrossMPT limits its potential for throughput improvement. To address this, a pipelining approach—commonly employed in various ECC decoders [1], [2]—can be applied to maximize CrossMPT's decoding throughput (see Figure 3 in [1]). By unrolling two cross-attention blocks, CrossMPT can simultaneously process two consecutive codewords across two cross-attention blocks within the same layer. This means that while the second cross-attention block processes the first codeword, the first cross-attention block concurrently decodes the subsequent codeword. This pipelining strategy ensures that CrossMPT maintains speed advantages over ECCT, in fully parallel scenarios. Figure 13 provides an example of decoding multiple codewords in CrossMPT with $N=2$.
> >
>
> All modifications are highlighted in the revised manuscript.
>
> ### References
>
> [1] M. Li et al., “High-speed LDPC decoders towards 1 Tb/s,” IEEE Transactions on Circuit and Systems I: Regular Papers, 2021.
>
> [2] M. Rowshan et al., “Channel coding toward 6G: Technical overview and outlook,” IEEE Open Journal of the Communications Society, 2024.

---

> > ### Comment · Reviewer_HjdF · 2024-11-27
> >
> > I thank the authors for their rebuttal.
> > I raised my score.

---

### Official Review · Reviewer_UA4Y · 2024-10-30

**Soundness:** 4
**Presentation:** 3
**Contribution:** 2
**Rating:** 6
**Confidence:** 5

**Summary:**

The paper proposes a new cross-attention based decoding schemes for linear block codes using transformer architecture. The masking and the cross attention mechanism depend on the parity check matrix of the block code, thus introducing the domain knowledge about the code into the decoding architecture. This work seems to be built on Error Correction Code Trabsformer (ECCT), NeurIPS 2022.

**Strengths:**

1. The idea of treating syndrome and magnitude as two separate entities and using cross-attention to capture the relation is novel and interesting. From a error correction point of view, this is more intuitive and understandable than existing approches.

2. The performance improvement over ECCT is compelling and non-trivial, while maintaining similar or lower complexity.

3. Overall well written and self-contained. Easy to understand and follow the ideas, even for reader unfamiliar with prior work.

**Weaknesses:**

1. One common troubling trend in comparing the performance of neural decoders is the choice of weak classical baselines. While I understand the difficulty of a common decoder architecture outperforming the highly specialised decoders for each of the class of codes, the comparison should neverthless be done for completeness. For instance, in Fig.4, only BP classical baseline is considered. It is known that BP is not optimal for many codes. For instance, when comparing with BCH codes, Berlekamp-Massey decoding algorithm should be used as the classical non-learning baseline and similarly for Polar codes, successive cancellation list decoding should be used. I strongly suggest authors to include the best classical baselines for each code to clearly show the gap with respect to the SOTA neural decoder, whether it is positive or negative.

2. I beleive authors chose the format of the negative log BER for presenting the results based on the original ECCT paper. But this format is not very informative and very uncommon in information and coding theory literature. While it is easy to identify which scheme is doing better, the gap between the performances is hard to interpret as the scale is not very intuitive. Rather, the standard format of SNR gap in dB for a target BER/BLER (ex. BLER target of $10^{-5}$) is more informative.

3. While the idea and results are interesting, the changes in the algorithm/architecture are only incremental compared to the original ECCT paper, which limits the novelty to some extent.

4. As with the case of other model-free neural decoders, there is a lack of interpretation for the neural decoder presented. Since the algorithmic contributions are limited, I would suggest atleast adding a discussion on interpretation and analysis, similar to the analysis and interpretation of neural belief propagation codes. A couple of references I can think of are: [1], [2]

[1] "Neural- network-optimized degree-specific weights for ldpc minsum decoding", Wang et. al. ISTC 2021.

[2] "Interpreting Neural Min-Sum Decoders", Ankireddy et. al, ICC 2023.

**Questions:**

Included in the weaknesses

---

> ### Author Response · Authors · 2024-11-21
> **Response**
>
> ## Comparison with SCL decoder and BM decoder
>
> Thank you for the insightful comments regarding code-specific decoders. As the reviewer mentioned, there are well-established decoding algorithms tailored for specific code classes. For instance, LDPC codes have the BP decoder, BCH codes employ the Berlekamp-Massey (BM) decoding algorithm, and polar codes rely on the SCL decoder.
>
> In our manuscript, we included comparisons of CrossMPT with these specialized decoders:
>
> - In **Appendix B**, we compared the decoding performance of CrossMPT with the BP decoder for LDPC codes.
> - In **Appendix D**, we evaluated CrossMPT against the SCL decoder for polar codes.
>
> For BCH codes, we conducted additional simulations, and the table below presents a performance comparison between the BM decoder, ECCT, and CrossMPT. The results demonstrate that CrossMPT outperforms both the BM decoder and ECCT.
>
> A key advantage of ECCT and CrossMPT is its versatility. While conventional decoders are good and valid only for respective code classes, CrossMPT, as a transformer-based universal decoder, performs effectively across a wide range of code classes. This universality highlights the broader applicability and potential of CrossMPT in various decoding scenarios and future communication paradigm such and semantic communication.
>
> | **(31,16) BCH** | **1 dB** | **2 dB** | **3 dB** | **4 dB** | **5 dB** | **6 dB** |
> | --- | --- | --- | --- | --- | --- | --- |
> | BM | 9.58E-02 | 5.98E-02 | 3.02E-02 | 1.16E-02 | 3.14E-03 | 5.49E-04 |
> | ECCT | 5.70E-02 | 2.48E-02 | 7.95E-03 | 1.74E-03 | 2.59E-04 | 2.49E-05 |
> | CrossMPT | 5.01E-02 | 1.97E-02 | 5.30E-03 | 8.93E-04 | 8.93E-05 | 4.30E-06 |
> | **(63,36) BCH** | **1 dB** | **2 dB** | **3 dB** | **4 dB** | **5 dB** | **6 dB** |
> | BM | 9.86E-02 | 5.86E-02 | 2.49E-02 | 6.66E-03 | 9.17E-04 | 5.91E-05 |
> | ECCT | 9.62E-02 | 5.53E-02 | 2.32E-02 | 6.50E-03 | 1.01E-03 | 8.08E-05 |
> | CrossMPT | 9.46E-02 | 5.44E-02 | 2.29E-02 | 6.46E-03 | 1.02E-03 | 7.62E-05 |
> | **(63,45) BCH** | **1 dB** | **2 dB** | **3 dB** | **4 dB** | **5 dB** | **6 dB** |
> | BM | 8.33E-02 | 5.19E-02 | 2.47E-02 | 7.80E-03 | 1.46E-03 | 1.42E-04 |
> | ECCT | 7.87E-02 | 4.36E-02 | 1.67E-02 | 3.69E-03 | 4.24E-04 | 1.95E-05 |
> | CrossMPT | 7.62E-02 | 4.03E-02 | 1.42E-02 | 2.75E-03 | 2.63E-04 | 1.01E-05 |
> | **(63,51) BCH** | **1 dB** | **2 dB** | **3 dB** | **4 dB** | **5 dB** | **6 dB** |
> | BM | 7.34E-02 | 4.67E-02 | 2.38E-02 | 8.67E-03 | 1.98E-03 | 2.61E-04 |
> | ECCT | 7.17E-02 | 4.07E-02 | 1.58E-02 | 3.59E-03 | 3.68E-04 | 1.62E-05 |
> | CrossMPT | 7.09E-02 | 3.93E-02 | 1.47E-02 | 3.11E-03 | 2.99E-04 | 1.29E-05 |
>
>
>
> ## SNR gap between ECCT and CrossMPT
>
> First of all, we totally agree with the reviewer that, rather than using the format of the negative log BER for presenting the result, BER/BLER are much popular and important metrics when evaluating decoder performance. We chose to report negative log BER in our evaluation because all related works on transformer-based decoders present their performance in this format [1], [2], [3], [4]. This convention is primarily due to page limitations, as it allows for a more concise presentation of results. Due to the page limit, we included some performance graphs in Appendices A, B, and C.
>
> To reflect the reviewer's feedback, we provide the standard format of SNR gain in dB for a target BER in the following table and we would like to include additional performance graphs in Appendix.
>
> | Codes | Target BER | SNR gap |
> | --- | --- | --- |
> | (31,16) BCH | $10^{-6}$ | 0.6 dB |
> | (128,86) Polar | $10^{-5}$ | 0.66 dB |
> | (128,64) CCSDS | $10^{-6}$ | 0.33 dB |
>
> The BER graphs including the SNR gap are shown in:
>
> - (31,16) BCH code: https://ibb.co/LhJ8kjx
> - (128,86) polar code: https://ibb.co/Jk915h1
> - (128,64) CCSDS: https://ibb.co/Cv7L1dd

---

> ### Author Response · Authors · 2024-11-21
> **Response-2**
>
> ## Novelty of CrossMPT
>
> Thank you for your valuable comments. While CrossMPT builds on the foundational  principles of ECCT, it introduces a novel decoding architecture that integrates a cross-attention mechanism and iterative decoding to address inefficiencies in ECCT. This approach provide concrete technical advancements compared to prior transformer-based decoders [1], [2], [3], [4], [5], which entirely rely on self-attention mechanisms.
>
> The incorporation of cross-attention mechanism is a key contribution to improve both decoding performance and decoding complexity. Notably, the proposed architecture aligns closely with the operational principles with well-established message-passing algorithms (e.g., sum-product and min-sum algorithms) in the field of classical channel coding. This alignment opens up a new direction for transformer-based decoders by leveraging the advanced modern techniques of message-passing algorithms, such as message update strategies, theoretical analysis, complexity-reduction techniques, and more. For example, the attention maps in Figures 5 and 6 show that CrossMPT’s attention map is directly connected to the parity-check matrix (PCM) of the underlying code. This connection could enable us to analyze the relationship between the PCM and the behavior of the transformer, providing a foundation for theoretical analysis as a further work.
>
>
> ## Interpretation and analysis of model-free decoder
>
> For model-based neural decoders, interpretation and analysis in terms of graphs are feasible because their structural architecture is inherently based on conventional graph-based decoding algorithms. For instance, as highlighted in the works you suggested [5] and [6], the degree of nodes and the cycles in the graph are closely related to the weight distribution in model-based neural decoders. However, in the case of model-free approaches, it remains challenging to determine how attention scores or weights are assigned to specific nodes and how these assignments are influenced by graph properties such as node degree.
>
> Nevertheless, in this study, we proposed a new interpretation of the attention score map due to CrossMPT's compact architecture. For example, as investigated in Figure 6 and Figure 11, we analyzed how the attention scores change depending on where the error occurs.
>
> However, we have not yet achieved a rigorous analysis beyond this level. As the reviewer pointed out, this remains a critical issue in model-free approaches and represents a problem that needs to be addressed in future work. Therefore, we plan to highlight this issue as a future research direction in the conclusion.
>
> ### References
>
> [1] Choukroun, Y. and Wolf, L. Error correction code transformer. *NeurIPS*, 2022.
>
> [2] Choukroun, Y. and Wolf, L. A foundation model for error correction codes. *ICLR*, 2024.
>
> [3] Choukroun, Y. and Wolf, L. Denoising diffusion error correction codes. *ICLR*, 2023.
>
> [4] Choukroun, Y. and Wolf, L. Learning linear block error correction codes. *ICML*, 2024.
>
> [5] Wang, L. et al. Neural- network-optimized degree-specific weights for ldpc minsum decoding. *ISTC*, 2021.
>
> [6] Ankireddy, SK. et al. Interpreting Neural Min-Sum Decoders, *ICC*, 2023.

---

> > ### Comment · Reviewer_UA4Y · 2024-11-23
> > **Response to Rebuttal**
> >
> > I thank the authors for their responses. I believe that many of my concerns have been addressed here and I increased my score accordingly.

---

### Official Review · Reviewer_kBMv · 2024-10-30

**Soundness:** 3
**Presentation:** 3
**Contribution:** 3
**Rating:** 8
**Confidence:** 5

**Summary:**

The paper proposes cross-attention-based message-passing transformers (CrossMPT), which solve the ambiguous masking problem of the naïve self-attention-based approach (ECCT). The magnitude and syndrome information are processed using two cross-attention layers, similar to message-passing algorithms, incorporating a mask defined by the parity check matrix. The proposed scheme achieves superior performance on various codes and SNRs consistently over ECCT and other baselines. The cross-attention map provides some degree of interpretability, while cross-attention itself enjoys reduced complexity compared to full self-attention, enabling the processing of long codes.

**Strengths:**

1. The update in the attention layer built upon ECCT is well-motivated and shows meaningful performance improvement. Although cross-attention is a well-known technique in the community, it is placed at the right position to improve the performance and efficiency of the previous self-attention-based approach.
2. The paper is easy to follow and provides sufficient details.
3. The experimental results effectively support the claims.
4. The in-depth analysis of the proposed method helps readers understand the paper better.

**Weaknesses:**

Although I think the paper is strong, I found a couple of minor weaknesses.

1. The actual inference time reduction from ECCT is not as great as the FLOPs improvement. Also, runtime evaluation and complexity analysis need more details about the environment. See Questions 3-5.
2. Related works on cross attention in other domains are missing, e.g., language (Vaswani et al., 2017), vision (Chen et al., 2021), text-based image generation (Rombach et al., 2022), etc.

(Vaswani et al., 2017) Vaswani, A. "Attention is all you need." *Advances in Neural Information Processing Systems* (2017).

(Chen et al., 2021) Chen, Chun-Fu Richard, Quanfu Fan, and Rameswar Panda. "Crossvit: Cross-attention multi-scale vision transformer for image classification." *Proceedings of the IEEE/CVF international conference on computer vision*. 2021.

(Rombach et al., 2022) Rombach, Robin, et al. "High-resolution image synthesis with latent diffusion models." *Proceedings of the IEEE/CVF conference on computer vision and pattern recognition*. 2022.

**Questions:**

1. Why was the magnitude chosen instead of another quantity, such as LLR? What is the intuition behind using the magnitude?
2. What happens if the syndrome information is placed as the query in the first attention layer? Does changing the order make any difference?
3. Is the inference time in Table 2 evaluated on the GPU?
4. What is the peak memory requirement?
5. Was the inference runtime evaluation conducted after code compilation (e.g., using `torch.compile()`)?

---

> ### Author Response · Authors · 2024-11-20
> **Response**
>
> ## Related works using cross-attention module in other domains
>
> Thank you for the recommendation for the related works. According to the reviewer’s comment, we would include the following references [1], [2], [3] in our related works employing cross-attention module.
>
> ## Reason for choosing magnitude instead of signed LLR
>
> In model-free decoders, the use of the magnitude instead of the ***signed*** LLR is intended to address the overfitting problem by employing the syndrome-based approach.
>
> In [4], the overfitting issue in model-free neural decoders is defined as follows:
>
> > *“A major challenge facing applications of deep neural networks to decoding is the avoidance of **overfitting the codewords** encountered during training.  Specifically,  training data is typically produced by randomly selecting codewords and simulating the channel transitions. Due to the large number of codewords (exponential in the block length), it is impossible to account for even a small fraction of them during training, leading to poor generalization of the network to new codewords. This issue was a major obstacle in [4] and [5],  constraining their networks to very short block lengths.”*
> >
>
> The syndrome-based approach [4], which utilizes the magnitude of received words and syndrome values to learn the channel noise, is theoretically proven to ensure that the decoder's performance is invariant to the codeword used for training [Theorem 1, 4]. Thus, when the signed LLR is used, the model-free decoders suffer from a severe overfitting problem.
>
> Otherwise, if the absolute values of LLRs are used, the results remain unchanged, as these values are merely a scaled version of the magnitude.
>
> ## Reversing the order of magnitude and syndrome update
>
> We reverse the order of magnitude and syndrome update of CrossMPT by using syndrome as the query in the first attention layer and magnitude as the query in the second attention layer. The following table compares the two methods: Updating magnitude first and then syndrome (denoted as M→S), and updating syndrome and then magnitude (denoted as S→M).  As shown in the table, the order of updating does not significantly impact the decoding performance, and both methods have almost the same decoding performance.
>
> | Method |  | M$\rightarrow$S |  |  | S$\rightarrow$M |  |
> | --- | --- | --- | --- | --- | --- | --- |
> | SNR | 4 dB | 5 dB | 6 dB | 4 dB | 5 dB | 6 dB |
> | (31,16) BCH | 6.98 | 9.25 | 12.48 | 7.00 | 9.33 | 12.40 |
> | (63,36) BCH | 5.03 | 6.91 | 9.37 | 5.09 | 6.93 | 9.58 |
> | (63,45) BCH | 5.90 | 8.20 | 11.62 | 5.92 | 8.24 | 11.74 |
> | (63,51) BCH | 5.78 | 8.08 | 11.41 | 5.76 | 8.04 | 11.34 |
> | (64,32) Polar | 7.50 | 9.97 | 13.31 | 7.58 | 10.09 | 13.23 |
> | (64,48) Polar | 6.51 | 8.70 | 11.31 | 6.50 | 8.58 | 11.10 |
> | (49,24) LDPC | 6.68 | 9.52 | 13.19 | 6.66 | 9.43 | 13.58 |
> | (121,60) LDPC | 5.74 | 9.26 | 14.78 | 5.74 | 9.28 | 14.92 |
> | (121,70) LDPC | 7.06 | 11.39 | 17.52 | 7.07 | 11.38 | 17.42 |
> | (96,48) MACKAY | 7.97 | 11.77 | 15.52 | 7.79 | 11.59 | 15.19 |

---

> ### Author Response · Authors · 2024-11-20
> **Response-2**
>
> ## Complexity analysis and the inference time
>
> All simulations were conducted on the NVIDIA GeForce RTX 3090 GPU, and the inference runtime evaluation was performed without code compilation.
>
> The FLOPs reported in Table 2 of the paper only accounted for the attention modules—two cross-attention modules for CrossMPT and one self-attention module for ECCT. However, since the total FLOPs are more critical for a comprehensive comparison, we have recalculated and presented the **total FLOPs** in the updated table below.
>
> | Codes | Method | FLOPs | Inference | training | Peak memory |
> | --- | --- | --- | --- | --- | --- |
> | (63,45) BCH | ECCT | 106.4 M | 759 us | 29 s | 1828 MiB |
> |  | CrossMPT | 99.8 M | 752 us | 29 s | 962 MiB |
> | (121,70) LDPC | ECCT | 256.8 M | 901 us | 80 s | 3926 MiB |
> |  | CrossMPT | 229.7 M | 844 us | 58 s | 1980 MiB |
> | (121,80) LDPC | ECCT | 238.0 M | 919 us | 76 s | 3602 MiB |
> |  | CrossMPT | 212.5 M | 849 us | 53 s | 1936 MiB |
> | (132,40) Turbo | ECCT | 343.4 M | 1009 us | 110 s | 5580 MiB |
> |  | CrossMPT | 303.6 M | 949 us | 83 s | 2362 MiB |
> | (255,223) BCH | ECCT | 53.5 M | 859 us | 145 s | 7318 MiB |
> |  | CrossMPT | 28.2 M | 747 us | 56 s | 1036 MiB |
> | (384,320) WRAN | ECCT | 111.3 M | 1553 us | 305 s | 18192 MiB |
> |  | CrossMPT | 53.1 M | 1189 us | 104 s | 3270 MiB |
>
> Additionally, we have included the peak memory usage during inference, where CrossMPT demonstrates a significant advantage over ECCT. All results are measured with $N=6,d=32$ for (255,233) BCH code, (384,320) WRAN and with $N=6,d=128$ for the others. This improvement arises from the reduced size of CrossMPT's attention map: as described in the manuscript, CrossMPT's attention map size is $2n(n-k$) (sum of two cross-attention modules), which is at most half the size of ECCT's attention map, $(2n-k)^2$. This leads to a significant reduction in memory usage as shown in the table. For example, the memory usage of ECCT for (255,233) BCH code is 7 times larger than that of CrossMPT.
>
> | Codes | CrossMPT | ECCT |
> | --- | --- | --- |
> | (63,45) BCH | 962 MiB | 1828 MiB |
> | (121,70) LDPC | 1980 MiB | 3926 MiB |
> | (121,80) LDPC | 1936 MiB | 3602 MiB |
> | (132,40) LTE Turbo | 2362 MiB | 5580 MiB |
> | (255,223) BCH | 1036 MiB | 7318 MiB |
> | (384,320) WRAN | 3270 MiB | 18192 MiB |
>
> During **training**, operations are typically **compute-bound**, meaning performance is primarily limited by the computational capacity of the hardware (e.g., FLOPs). As a result, reducing FLOPs significantly impacts training time.
>
> In contrast, **inference** is often **memory-bound**, where performance is constrained by memory bandwidth rather than computational power. This means that reducing FLOPs has less impact on inference time. For instance, in memory-bound scenarios, such as small batch inference, GPUs may spend more time fetching data than performing computations (e.g., for Llama2-70B, increasing GPUs from 4x to 8x reduces latency by only 0.7x).
>
> This difference arises due to several factors such as:
>
> 1. **Inference Optimization**: Inference dose not involve backpropagation and gradient calculations, reducing computational demands.
> 2. **Hardware Parallelism**: Inference takes advantage of GPU parallelism to effectively manage computational loads.
>
> As noted in ShuffleNet v2 [5], FLOPs alone are not sufficient to accurately represent real-world performance, as they fail to account for critical factors such as memory access patterns, hardware-specific characteristics, and the degree of parallelism. While FLOPs provide a useful estimate of computational load, other factors often dominate practical performance, particularly during inference.
>
> ### References
>
> [1] Vaswani, A. et al., Attention is all you need. *NeurIPS,* 2017.
>
> [2] Chen, C.-F. R. et al. "CrossVit: Cross-attention multi-scale vision transformer for image classification." *ICCV,* 2021.
>
> [3] Rombach, R. et al., "High-resolution image synthesis with latent diffusion models." *CVPR,* 2022.
>
> [4] Bennatan, A., Choukroun, Y., and Kisilev, P. “Deep learning for decoding of linear codes-a syndrome-based approach.” in *Proc. ISIT*, 2018.
>
> [5] Ma, N. et al., “Shufflenet v2: Practical guidelines for efficient CNN architecture design.” *ECCV*, 2018.

---

> > ### Comment · Reviewer_kBMv · 2024-11-25
> >
> > Thank you for the detailed response and for providing the additional results. They satisfactorily address all of my concerns. The peak memory reduction, which is one of the paper's major claims, appears significant and beneficial for operating the learning-based decoder on resource-constrained systems. Therefore, I am inclined to raise my score.

---

> ### Author Response · Authors · 2024-11-26
> **Official Comment by Authors**
>
> Thank you again for your questions and feedback, which helped us improve the manuscript, especially regarding peak memory reduction.
>
> We have carefully considered all your concerns and have uploaded the revised manuscript accordingly. All the modifications are highlighted in the revised manuscript.

---

### Official Review · Reviewer_VWDs · 2024-11-02

**Soundness:** 3
**Presentation:** 3
**Contribution:** 3
**Rating:** 6
**Confidence:** 5

**Summary:**

This paper studies the neural network based decoding of conventional error correction codes. In essence, decoding of an error correction code is a classification problem. Conventional belief propagation based decoders rely on the code structure to iteratively improve their beliefs on the message bits. An alternative approach is to parameterize the decoder as a neural network, and train it on various codewords. Given the large number of codewords, it is challenging to design such a decoder without exploiting the code structure. An obvious approach is to use the code structure and the basic steps of the belief propagation decoder, but unroll the iterations on the layers of a neural network. This work follows the recently introduced ECCT decoder in this direction, and exploits a transformer architecture for the decoder. In particular, by transforming the decoding operation into the estimation of the sign of the multiplicative noise component, and by separately representing the magnitude of the received signal and its syndrome as two separate pieces of information, the authors train separate masked cross-attention blocks to update the embeddings of these two blocks before reaching the final decision. Through experiments, they show that the resultant decoder can achieve a lower bit error rate (BER) compared to ECCT in decoding a number of different channel codes, with a slightly lower computational complexity.

**Strengths:**

The decoding of error correction codes is an important problem with significant potential impact in practice. Using neural network based decoders is a promising approach that has been receiving significant attention recently. The paper improves the state of the art transformer-based architecture with a simple yet effective enhancement in the architecture. Given that the magnitude and the syndrome provide different types of information on the transmitted symbols, it makes sense to process them separately, and learn different cross-attention modules. Despite the fact that the architectural modification is simple, the improvement in BER performance is significant. The authors also provide a good explanation of their motivation and the impact of their architecture on the decoding process (e.g., attention scores). Overall, it is a well-written paper with clearly highlighted contributions.

**Weaknesses:**

The architectural modification proposed in the paper with respect to ECCT is rather trivial. In that sense, it does not have significant technical novelty, on the other hand, given its impact on the performance, I believe it is a worthy contribution.

The paper presents only BER performance results. It is not clear if the proposed approach actually provides any gains in terms of the block error probability, which would really matter for the code.

**Questions:**

- The performance measure presented in the paper is BER; however, what really matters for a channel code is the block error probability (BLER). Therefore, I would ask the authors to include BLER results to clearly show that the proposed approach does actually improve the performance with respect to ECCT or other baseline decoding algorithms. Improvements in BER are still important, but correct decoding of the message bits would still require the concatenation of the code with an outer code, which would further increase the decoding complexity, and would diminish the gains of the proposed approach. I believe, not presenting the BLER results at all is not fair, and does not show the real performance of the approach.

- How does the complexity compare with the more conventional BP decoders and neural BP (NBP) decoders? Are the numerical results presented in Fig. 4 for the same complexity decoders, or for the same number of decoder iterations? If it's the latter, the comparison is not fair. If each iteration of the NBP decoder has half the complexity of the proposed approach, it can run twice the number of iterations, and potentially achieve lower BER.

- Why is the double-masked ECCT not included in the comparisons?

---

> ### Author Response · Authors · 2024-11-20
> **Response**
>
> ## Novelty of CrossMPT
>
> Thank you for your valuable comments. While CrossMPT builds on the foundational  principles of ECCT, it introduces a novel decoding architecture that integrates a cross-attention mechanism and iterative decoding to address inefficiencies in ECCT. This approach provide concrete technical advancements compared to prior transformer-based decoders [1], [2], [3], [4], [5], which entirely rely on self-attention mechanisms.
>
> The incorporation of cross-attention mechanism is a key contribution to improve both decoding performance and decoding complexity. Notably, the proposed architecture aligns closely with the operational principles with well-established message-passing algorithms (e.g., sum-product and min-sum algorithms) in the field of classical channel coding. This alignment opens up a new direction for transformer-based decoders by leveraging the advanced modern techniques of message-passing algorithms, such as message update strategies, theoretical analysis, complexity-reduction techniques, and more. For example, the attention maps in Figures 5 and 6 show that CrossMPT’s attention map is directly connected to the parity-check matrix (PCM) of the underlying code. This connection could enable us to analyze the relationship between the PCM and the behavior of the transformer, providing a foundation for theoretical analysis as a further work.
>
> ## BLER results for CrossMPT
>
> First of all, the reason why we chose to report BER in our evaluation is that all related works on transformer-based decoders demonstrated their performance in BER (negative log BER) [1], [2], [3], [4].
>
> Although we follow the convention of the prior papers, we agree that BLER (FER) is an important metric. In the following table, we present the BLER results, which, similar to the BER performance, demonstrate the superiority of CrossMPT.
> | Method |  | ECCT |  |  | CrossMPT |  |
> | --- | --- | --- | --- | --- | --- | --- |
> | SNR | 4 dB | 5 dB | 6 dB | 4 dB | 5 dB | 6 dB |
> | (31,16)   BCH | 1.52E-02 | 2.53E-03 | 2.86E-04 | 6.00.E-03 | 7.15.E-04 | 3.33.E-05 |
> | (63,36)   BCH | 8.77E-02 | 1.66E-02 | 1.66E-03 | 8.19.E-02 | 1.45.E-02 | 1.35.E-03 |
> | (63,45)   BCH | 6.41E-02 | 8.65E-03 | 4.68E-04 | 4.14.E-02 | 4.82.E-03 | 1.74.E-04 |
> | (63,51)   BCH | 6.58E-02 | 7.83E-03 | 4.00E-04 | 5.30.E-02 | 5.72.E-03 | 2.33.E-04 |
> | (64,32) Polar | 1.53E-02 | 1.55E-03 | 1.15E-04 | 7.96.E-03 | 7.05.E-04 | 3.00.E-05 |
> | (64,48) Polar | 4.58E-02 | 6.34E-03 | 4.98E-04 | 2.84.E-02 | 3.41.E-03 | 2.36.E-04 |
> | (49,24) LDPC | 2.67E-02 | 2.76E-03 | 8.39E-05 | 1.14.E-02 | 7.47.E-04 | 2.31.E-05 |
> | (121,70) LDPC | 4.02E-02 | 1.24E-03 | 6.69E-06 | 1.38.E-02 | 2.21.E-04 | 1.59.E-06 |
>
>
> ## Comparison with conventional BP and NBP
>
> First, we would like to clarify that the results in Table 1 and Fig. 4 are derived from the original ECCT work [1]. As the reviewer pointed out, conventional BP and NBP decoders indeed have lower complexity compared to transformer-based decoders. However, a key consideration is the decoding performance upon convergence.
>
> For example, in Appendix B, we compared the decoding performance of CrossMPT with that of the conventional BP decoder for maximum iterations of 20, 50, and 100. The results show that the BP decoder converges sufficiently within approximately 100 iterations, meaning that increasing the number of iterations beyond this point does not provide additional performance improvement.
>
> In contrast, CrossMPT outperforms BP decoders in terms of best decoding performance, highlighting its advantage despite its higher complexity. This demonstrates the significance of CrossMPT’s superior performance, as it achieves greater reliability even where BP decoders reach their performance limits.
>
> The concise comparison in terms of complexity is currently challenging because the operations used in BP decoders and transformer-based decoders differ significantly (e.g., BP decoders rely on sum, tanh, and arctanh functions, while transformer-based decoders use attention mechanisms and matrix multiplications). Such complexity analysis requires further in-depth research.
>
> It is important to note that the objective of this study is not to propose a transformer-based decoder as a replacement for BP decoders but to present an improved architecture for existing transformer-based decoders. Transformer-based decoders have a unique universality, as they operate independently of specific code classes, distinguishing them from traditional code-specific decoders. This universality highlights the necessity for their continued development and refinement.

---

> > ### Comment · Reviewer_VWDs · 2024-11-26
> >
> > Thank you for your detailed response. I appreciate the BLER results and the arguments regarding the comparison with BP and NBP decoders. Again, I believe the paper makes a worthy contribution to transformer based decoding approach; however, just saying that the focus is improving such codes is not sufficient on its own. If transformer-based codes perform clearly worse than existing codes, with higher computational complexity, this may limit the value of all such works, even if it improves the performance of codes within that class. I still value results in this direction as these issues may be resolved with further research, but this needs to be made clear in the paper.
> >
> > From the numbers presented for BLER, it seems to me that the performance is generally worse than state of the art decoders. Can you please include those in your comparison, and also add the BLER figure to the paper? I think it should be in the main part of the paper as it’s an essential performance measure for a code, but if this will be difficult due to space constraints, please include the comparison table and the accompanying discussion at least in an Appendix.
> >
> > I believe my original score was already matching the contribution of the paper; and therefore, I will retain it.

---

> > > ### Author Response · Authors · 2024-11-26
> > >
> > > Thank you for the comments and we are able to improve our manuscript according to your comments. We include the table with BLER results for various code classes and BLER performance graphs for (31,16) BCH code, (63,51) BCH code, and (648,540) IEEE 802.11n LDPC code in **Appendix D** of the revised manuscript.
> > >
> > > For BCH codes, we compare the decoding performance of CrossMPT with the traditional Berlekamp-Massey (BM) decoder, maximum likelihood (ML) decoding algorithm, ECCT.
> > > For LDPC codes, we compare the decoding performance of CrossMPT with the traditional BP decoder with a maximum number of iterations of 20, 50, and 100.
> > > As shown in the table, CrossMPT outperforms ECCT in the BLER performance and also has a comparable BLER results compared to the traditional decoding algorithms.
> > >
> > > In addition, the traditional decoders are code-specific decoders, tailored to each class of codes. For example, LDPC codes are effectively decoded by the BP decoder, BCH codes by the BM decoder, and polar codes by the SCL decoder. However, unlike the traditional decoders, a key advantage of CrossMPT is its versatility. While conventional decoders are good and valid only for respective code classes, CrossMPT performs effectively across a wide range of code classes. This universality highlights the broader applicability and potential of CrossMPT in various decoding scenarios and future communication paradigm such and semantic communication.
> > >
> > > The complexity of transformer-based decoders is relatively high compared to code-specific decoders. Reducing their computational requirements will be an important focus for future work.

---

> ### Author Response · Authors · 2024-11-20
> **Response-2**
>
> ## Applying double-masked structure to CrossMPT
>
> Double-masked (DM) ECCT [5] leverages the fact that different parity check matrices of the same code provide different performance. It utilized two different PCMs for the same linear code to capture the diverse relationships. Thus, the approach of double-masked ECCT is entirely different from CrossMPT, which introduces a new transformer architecture. Importantly, the double-masked structure of DM ECCT can also be applied to CrossMPT. The following table presents the results of DM ECCT, CrossMPT, and DM CrossMPT (which incorporates the double-masked structure into CrossMPT) for (31,16) BCH code and (63,45) BCH code with $N=6,d=128$. As shown in the following table, the vanilla CrossMPT outperforms DM ECCT. Notably, CrossMPT achieves nearly the same performance as DM ECCT despite utilizing only about half the number of parameters. Moreover, applying the double-masked structure to CrossMPT (DM CrossMPT) further improves its decoding performance.
>
> | Method |  | DM ECCT |  |  | CrossMPT |  |  | DM+CrossMPT |  |
> | --- | --- | --- | --- | --- | --- | --- | --- | --- | --- |
> | SNR | 4 dB | 5 dB | 6 dB | 4 dB | 5 dB | 6 dB | | 4 dB | 5 dB | 6 dB |
> | (31,16) BCH | 1.07.E-03 | 8.35.E-05 | 5.75.E-06 | 9.26.E-04 | 9.63.E-05 | 3.79.E-06 | | 9.05.E-04 | 8.72.E-05 | 3.81.E-06 |
> | (63,45) BCH | 2.74.E-03 | 1.18.E-03 | 9.31.E-06 | 2.74.E-03 | 2.74.E-04 | 9.01.E-06 | | 2.40.E-03 | 2.21.E-04 | 6.68.E-06 |
>
> ## References
>
> [1] Choukroun, Y. and Wolf, L. Error correction code transformer. *NeurIPS*, 2022.
>
> [2] Choukroun, Y. and Wolf, L. A foundation model for error correction codes. *ICLR*, 2024.
>
> [3] Choukroun, Y. and Wolf, L. Denoising diffusion error correction codes. *ICLR*, 2023.
>
> [4] Choukroun, Y. and Wolf, L. Learning linear block error correction codes. *ICML*, 2024.
>
> [5] S.-J. Park et al. How to mask in error correction code transformer: Systematic and double masking. arXiv, 2023.

---

### Official Review · Reviewer_caLP · 2024-11-02

**Soundness:** 4
**Presentation:** 3
**Contribution:** 4
**Rating:** 6
**Confidence:** 4

**Summary:**

This paper introduces a novel transformer-based architecture called CrossMPT(Cross-attention Message-Passing Transformer) for decoding error-correcting codes(ECCs). The key innovation lies in the use of cross-attention mechanisms to separately and iteratively update magnitude and syndrome vectors, in contrast to conventional transformer-based decodes that employ self-attention without distinguishing between different types of input vectors. The mask matrices in CrossMPT are derived from the code’s parity-check matrix (PCM), explicitly capturing the relationships between magnitude and syndrome vectors. Results demonstrate that CrossMPT significantly outperforms existing neural network-based decoders across various code classes, while also reducing memory usage, computational complexity, inference time, and training time.

**Strengths:**

1. Innovative architecture: introduced the cross-attention mechanisms to separately handle magnitude and syndrome vectors is a notable advancement, bridging concepts from traditional message-passing algorithms and modern transformer architectures.
2. Performance Gains Significantly: in the experiments show significant improvements to existing decoders across various code classes, including BCH, polar, LDPC, and turbo codes.
3. Efficiency Improvements: By reducing memory usage and computational complexity, CrossMPT addresses practical concerns associated with deploying transformer-based decoders, especially for longer codes.
4. Use of PCM-derived Masks: Aligning the mask matrices with the PCM directly incorporates code-specific structural information into the model, enhancing its ability to learn relevant relationships.

**Weaknesses:**

1. Availability of Code or Experimental Details: The paper does not mention whether the code or experimental records will be made available. Open-sourcing the code or providing detailed experimental records are crucial for reproducibility and for the community to gain more benefit from this work.

2. Code length: In the appendix H of "A Foundation Model for Error Correction Codes," the authors compare Maximum Likelihood decoders on the BCH(1023,1013) code. In this paper, authors highlight the significant reductions in memory usage. It would be valuable for the authors to extend this analysis by demonstrating performance on even longer codes or comparing their results against other decoding methods for these extended lengths. This could provide clearer insights into the advantages of their approach.

**Questions:**

1. FER Results: In [1], the authors present their results using the Frame Error Rate (FER) metric, while your results are reported using Bit Error Rate (BER). Could you please explain why you chose to report BER over FER in your evaluation? Are there specific challenges or reasons that prevent you from providing FER results for your approach? Including FER results would be helpful for assessing the practical decoding performance and for making a direct comparison with the results in [1].

2. Training Convergence and Speedup: In Appendix I, authors mention that CrossMPT achieves faster training convergence compared to ECCT. However, the figure provided does not clearly demonstrate a significant speed up. Could you please provide specific data quantifying how much faster CrossMPT converges during training? For instance, you could report metrics such as: wall-clock training time taken by each model to achieve comparable performance levels and percentage improvement in convergence speed or reduction in epochs/time needed.

3. Reproducibility: Do you plan to release the source code or detailed experimental records? Reproducibility is essential for validating the results and facilitating future research based on your work.

[1] A Foundation Model for Error Correction Codes

---

### Meta-Review · Area_Chair_gjBN · 2024-12-15

**Metareview:**

The paper builds on the existing ECCT approach for error correction via transformers, and it proposes a novel method dubbed cross-attention-based message-passing transformers (CrossMPT). The idea is to process magnitude and syndrome information using two cross-attention layers, incorporating a mask defined by the parity check matrix. The superiority over ECCT is demonstrated in a series of rather thorough experiments on various codes and SNRs.

The reviewers praised the innovative architecture, which leads to performance and efficiency improvements. Most of the issues raised in the initial reviews have been successfulIy addressed in the rebuttal and discussion. One problem that remains is that the proposed approach is still not competitive w.r.t. state-of-the-art non-neural alternatives. However, this problem is common to virtually all transformer-based decoders and the paper does take a step in the right direction providing novel ideas and improvements. Therefore, I recommend acceptance of the paper.

**Additional Comments On Reviewer Discussion:**

The reviewers raised a number of points (about long codes, BLER results, comparison with conventional BP and NBP, comparison with SCL decoding) that were mostly addressed in a rather satisfactory way by the rebuttal and subsequent discussion.

---

### Decision · Program_Chairs · 2025-01-22

Accept (Poster)